JGP Journal of General Physiology

Voltage-Gated Na Channels 2026

# Long-term inactivation mediated by different FGF-A homologues on heterologously expressed Na$_V$1.2 currents

Yenisleidy Lorenzo-Ceballos[1], Pedro L. Martinez-Espinosa[1], Xiao-Ming Xia[1], and Christopher J. Lingle[1]

A-isoforms of cytosolic growth factor homologous factors (FGF11–14) mediate long-term inactivation (LTI) of voltage-dependent sodium (Na$_V$) channels. LTI is a rapid onset process that is competitive with the fast inactivation (I$_F$) intrinsic to Na$_V$ channels, with little or no interconversion between inactivated states. Since recovery from LTI is orders of magnitude slower than recovery from I$_F$, repetitive depolarizations lead to use-dependent accumulation of Na$_V$ channels in slow recovery states, thereby limiting Na$_V$ availability during trains of action potentials. Of the two or more N-terminal splice variants of the various FGF homologues, LTI specifically arises only from the A-isoform of each FGF subunit. Although there is substantial homology among the N termini of the four FGF-A paralogs, to what extent LTI generated by the different FGF-A homologues may differ has not been directly addressed. Here, using heterologous expression in HEK293T cells, we evaluate the kinetics of onset and recovery from LTI mediated by hFGF11–14A in association with WT hNa$_V$1.2. We also use Na$_V$ channels with fast inactivation removed (IQM) to measure rates of LTI-mediated inactivation and recovery in the absence of intrinsic fast inactivation. Among the four FGF-A homologues, we identify two features that can differ. First, different FGF-A's differ in the rate of onset into LTI. Second, the rate of recovery from inactivation, whether measured with WT Na$_V$1.2 or with Na$_V$1.2_IQM, differs among FGF-A's. The functional differences among FGF-A homologues differentially sculpt the time course and extent of use-dependent accumulation of Na$_V$1.2 channels into LTI. This, in turn, would differentially impact on Na$_V$ availability during repetitive firing.

## Introduction

Fast inactivation (I$_F$) that is intrinsic to the gating behavior of voltage-dependent sodium (Na$_V$) channels typically occurs with sufficiently rapid onset and recovery that, in many cells, it readily permits repetitive firing well in excess of 50 Hz with little attenuation in Na$_V$ availability (Bean, 2007). Historically, such inactivation has been linked to elements in the peptide chain that connects domains III (DIII) and IV (DIV) of the four homologous domains that contribute to Na$_V$ channels. A key element in the DIII–DIV linker is a triplet of hydrophobic residues, isoleucine-phenylalanine-methionine (IFM) (Patton et al., 1992; West et al., 1992), that is shared among virtually all Na$_V$ homologues. However, although it had been proposed that the IFM motif and DIII–DIV linker might directly mediate Na$_V$ channel occlusion by forming a so-called "hinged lid" (Ahern et al., 2016), recent evidence suggests that intrinsic fast inactivation may involve an allosteric change, perhaps linked to movement of the S4 voltage sensor in DIV (Capes et al., 2013), that leads to constrictions within the pore created by rings of hydrophobic residues (Liu et al., 2023).

Over the past 20 years, a distinct inactivation process, termed long-term inactivation (LTI), has been identified that can also produce fast inactivation of Na$_V$ channels (Laezza et al., 2009; Dover et al., 2010; Goldfarb, 2012; Barbosa et al., 2017). Rates of onset of LTI are similar to or only slightly slower than I$_F$-mediated inactivation (Dover et al., 2010; Martinez-Espinosa et al., 2021b), but recovery from LTI occurs at 10–100-fold slower rates (Martinez-Espinosa et al., 2021a; Martinez-Espinosa et al., 2021b). As a consequence, since I$_F$- and I$_{LTI}$-mediated inactivation are considered competitive processes, during repetitive depolarizations a slow increase in the fraction of Na$_V$ channels in LTI-inactivated states will occur. Thus, the differential entry into the two pathways and accumulation in slow recovery states will result in use-dependent changes in Na$_V$ availability during repetitive cell firing. The aim of the present work is to examine

[1]Department of Anesthesiology, Washington University School Medicine, St. Louis, MO, USA.

Correspondence to Christopher J. Lingle: clingle@morpheus.wustl.edu.

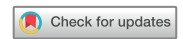

Table 1.  **FGF A-isoform N termini**

| Homologue | N-terminal residues | $f(LTI)$[a] |
|---|---|---|
| FGF11A (FHF3A) | M-AALASSLI RQKREVREPG GSRPVSAQRR VCPRGTKSLC QKQLLILLSK VRLCGGRPAR PDRGP EPQLKGIVTKL FCRQGFYLQA... | – |
| FGF12A (FHF1A) | MAAAIASSLI RQKRQARESN SDRVSASKRR SSPSKDGRSL CERHVLGVFS PVRRRP EPQLKGIVTRL FSQQGYFLQM... | ~0.36 |
| FGF13A (FHF2A) | MAAAIASSLI RQKRQARERE KSNACKCV–– SSPSKGKTSC DKNKLNVFSR VKLFGSKKRR RRRP EPQLKGIVTKL YSRQGYHLQL... | ~0.51 |
| FGF14A (FHF4A) | MAAAIASGLI RQKRQAREQH WDRPSASRRR SSPSKNRGLC NGNLVDIFSK VRIFGLKKRR LRRQ DPQLKGIVTRL YCRQGYYLQM... | ~0.58 |
| FGF14B (FHF4B) | MVKPVPLFRR TDFKLLLCNH KDLFFLRVSK LLDCFSPKSM WFLWNIFSKG THMLQCLCGK SLKKNKNPT DPQLKGIVTRL YCRQGYYLQM... | <0.1 |
| FGF14A-ΔNT | M– DPQLKGIVTRL YCRQGYYLQM... | |

[a]Approximate fractional diminution during four pulse train protocol with Na_V1.6 (Dover et al., 2010). Underlined residues correspond to exon 1; non-underlined corresponds to initial segment encoded by exon 2.

quantitatively the similarities and differences among fibroblast growth factor homologous factor (FGF)-A homologues in rates of onset and recovery from LTI to provide a framework for understanding potential impacts on use-dependent changes during repetitive depolarizations.

Given the slow recovery from inactivation characteristic of LTI, it is important to emphasize that LTI is considered a mechanistically distinct process from the intrinsic slow inactivation observed for most Na_V channels (Goldfarb, 2012). The primary difference is that LTI is a rapid onset process occurring within milliseconds, whereas slow inactivation is of slow onset developing over 100s of milliseconds or even seconds to minutes (Ulbricht, 2005).

FGFs are cytosolic, non-secreted proteins (but see Sochacka et al., 2020) and interact with the C- terminus of Na_V subunits (Goetz et al., 2009) in a region that may also be influenced by Na_V β subunits and calmodulin (Wang et al., 2012). Of the four FGF homologues (FGF11–14), FGF12 and 14 occur as either A- or B-isoforms; FGF11 only occurs as an A isoform; and FGF13 occurs in A, B, and VY isoforms (Yang et al., 2016). Functional comparisons of A- and B-isoforms of FGF12–14, when each are coexpressed in heterologous expression systems with Na_V1.6, have clearly shown that only the A-isoforms mediate LTI (Laezza et al., 2009; Dover et al., 2010). Furthermore, specific mutations within the first 20 N-terminal residues of FGF13A impede the occurrence of LTI (Dover et al., 2010). The N termini of all four FGF-A homologues share extensive amino acid identity through much of the N terminus (Mahling et al., 2021), but sequence and length differences do exist (Table 1). Whether there may be differences among these N termini that may result in isoform-dependent differences in LTI has not been adequately addressed. A commonly used qualitative test for the presence of LTI is application of a 4–10-pulse train of depolarizing stimuli applied at, say, 5 or 10 Hz. With an intact LTI mechanism, this results in substantial diminution of peak Na_V current amplitude. When FGF12, 13A, or 14A are coexpressed with Na_V1.6, there are suggestions that the steady-state level of LTI at a given stimulus frequency may differ among A-homologues (see Table 1 [Laezza et al., 2009; Dover et al., 2010]), but this requires closer attention.

Differences in rates of onset, recovery, and extent of LTI might be expected to underlie potentially important differences in use-dependent changes in Na_V availability among different combinations of Na_V channels and FGF-A homologues. Here, we compare specific rates of onset and recovery from inactivation among the four FGF-A homologues when coexpressed with the human Na_V1.2 (hNa_V1.2) channel. Our results reveal differences among both inactivation onset and recovery among the different FGF-A isoforms and show these differentially impact on use-dependent changes in Na_V availability. We use the results to provide further quantitative evaluation of the LTI framework used by Goldfarb and others (Dover et al., 2010). Overall, the simple competitive model between IFM and FGF-A mediated inactivation processes is generally consistent with most aspects of the results. However, we note that, when FGF-A homologues are coexpressed with Na_V1.2, the rate of onset of inactivation does not increase as much as predicted. We discuss this in regard to the possibility that the presence of an FGF subunit may allosterically slow transitions involved in the onset of intrinsic fast inactivation, thereby facilitating the fraction of channels that enter LTI.

## Materials and methods
### Constructs
During the course of this work, we utilized two different hNa_V1.2 constructs. One was kindly provided by Dr. Geoffrey Pitt (Columbia University, New York, NY, USA), to which an mCherry reporter was introduced downstream of the hNa_V1.2 reading frame. The second was obtained from Addgene (pIR-CMV-SCN2A-Variant-1-IRES-mScarlet, Cat #162279; Addgene). This construct (DeKeyser et al., 2021) had been deposited with Addgene by Dr. Al George (Northwestern University School of Medicine, Chicago, IL, USA). Both Na_V1.2 constructs were subsequently modified by site-directed mutagenesis. In one case, we introduced substitutions at predicted ubiquitination sites (Y1975A/V1978D; denoted Na_V1.2_{YV/AD}). In a second, we introduced a substitution in the IFM motif (F1489Q; Na_V1.2_{YV/AD}_IQM) to modify intrinsic fast inactivation. All experiments reported here utilized the Na_V1.2_{YV/AD} background.

FGF clones (FGF12-14, A and B isoforms; FGF11A) were kindly provided by Dr. Jeanne Nerbonne (Washington University School of Medicine, St. Louis, MO, USA) and subcloned into an pEYFP vector. The vector has two expression cassettes, one (CMV promoter/hGH PolyA) for controlling expression of the FGF gene and another (mPKG promoter/gGH PolyA) for EYFP fluorescent protein. N-terminal mutations of FGF14A were introduced into the FGF14A-EYFP plasmid by site-directed mutagenesis: LI/AA (L9A/I10A), 2Q (R17Q/W21Q) and 5Q (R11Q/K13Q/R14Q/R17Q/W21Q). The truncated FGF14A construct lacking N-terminal 64 residues (FGF14A_ΔNT) was generated by PCR reaction and subsequently cloned into a pEYFP vector.

To assist expression of $Na_V1.2$, in early experiments, we utilized $hNa_V\beta1$ (SCNB1) and $hNa_V\beta2$ (SCNB2) genes (kindly provided by Dr. Jon Silva, Washington University, St. Louis MO, USA), each of which were subcloned into vectors, also encoding pmEGFP or pECFP, to enable the separate expression of $Na_V\beta$ genes as well as the fluorescent proteins mEGFP or ECFP. However, for most experiments described here, we utilized an HEK293T cell line obtained from Dr. Al George (Northwestern University School of Medicine) that was stably transfected with $Na_V\beta1$ and $Na_V\beta2$ subunits (Kahlig et al., 2010; Thompson et al., 2023).

## Cell lines and transfection methods

Two sources of HEK293T cells were used in this work. In early experiments, HEK293T cells were obtained from ATCC (CRL-11268). In later experiments, we utilized the stably transfected cell line, described above, that expresses both $Na_V\beta1$ (SCNB1) and $Na_V\beta2$ (SCNB2) subunits (HEK293T-β1/β2) (Thompson et al., 2023). Cells were maintained in Dulbecco's modified Eagle medium (Thermo Fisher Scientific) supplemented with 10% fetal bovine serum (FBS), 100 U/ml penicillin, 100 μg/ml streptomycin, and 2 mM L-glutamine. For HEK293T-β1/β2 cells, the culture medium was additionally supplemented with 3 μg/ml puromycin. Cultures were kept at 37°C in a humidified incubator with 5% $CO_2$. For electrophysiology experiments, HEK cells (HEK293T or HEK293T-β1/β2) were transiently transfected with WT or IQM alone or co-transfected with each FGF construct using 293fectin following the manufacturer's instructions (Thermo Fisher Scientific). Briefly, cells were plated at ~50–60% confluence in 35-mm dishes at time of transfection. For single transfections, 3 μg total DNA of $Na_V1.2$ (WT or IQM) was used. For coexpression experiments, $Na_V1.2$ (WT or IQM) and each FGF construct were transfected at 1:1 DNA ratio (total 3 μg DNA). DNA was mixed with 293fectin at 1:2 (μg: μl) ratio in Opti-MEM medium (Thermo Fisher Scientific) and applied to the cultures. Approximately 30 h after transfection, cells were seeded at low density onto laminin/ploy-D-lysine–coated coverslips (Electron Microscopy Sciences), and cells were used for electrophysiological recordings 48 h after transfection.

Despite the use of fluorescent reporters to identify cells with likely channel expression, we encountered considerable variability in magnitude or even the presence of $Na_V$ current among fluorescent cells. For this analysis, we limited our analysis to cells with >500 pA of $Na_V$ current at the peak of the $Na_V$ IV curve. To provide an assessment of the potential occurrence of endogenous $Na_V$ currents that might confound evaluation of expressed $Na_V$, we also obtained whole-cell recordings from a set of 37 HEK293T cells transfected exclusively with the SCN1B vector. For this set of cells, the average (±SD) inward current density was −5.3 ± 3.3 pA/pF ($N = 37$), with a maximum in one cell of −20.54 pA. In comparison, for the set of cells for which IV curves were generated in this analysis, the peak current values were for WT $Na_V1.2$, −485.0 ± 380.2 pA/pF ($N = 14$), for $Na_V1.2$+FGF14A, −392.6 ± 193 pA/pF ($N = 15$); for $Na_V1.2$+FGF13A, −663.3 ± 357.5 pA/pF ($N = 5$); for $Na_V1.2$+FGF12A, −339.5 ± 290.9 pA/pF ($N = 5$); and for $Na_V1.2$+FGF11A, −434.6 ± 378.5 pA/pF ($N = 5$). For a set of 162 cells, membrane capacitance was 13.7 ± 5.4 pF with a minimum of 5.1 pF and maximum of 34 pF. Furthermore, the distinct functional properties conferred on the $Na_V$ currents by FGF subunits, particularly the kinetics of recovery from inactivation reported here, also provide confidence that the currents in this study arise exclusively from transfected subunits.

We do note one aspect of $Na_V1.2$+FGF13A currents not previously reported in earlier studies that may confound the assessments of fractional LTI during long duration recordings from some cells. Specifically, in our hands, currents arising from $Na_V1.2$ coexpression with FGF13A exhibited substantial time-dependent changes in fraction of channels that enter LTI following formation of the whole-cell recording. This varied among different cells. Therefore, data analyzed for $Na_V1.2$+FGF13A were generally limited to the first 10 min during which time SSI and recovery protocols could be acquired with little change in the fraction of channels entering LTI. However, longer protocols (e.g., Figs. 10 and 11) that tested use-dependent changes in LTI by examining the impact of pulse trains on subsequent LTI were vulnerable to the gradual diminution in fraction of channels entering LTI that occurred over recording time. As noted in the Results, tests of the extent to which $Na_V1.2$+FGF13A channels can be driven into LTI during longer recordings may likely be underestimates. Factors that may influence the time-dependent changes in LTI for FGF13A will be addressed elsewhere.

## Electrophysiology and data analysis

Transiently transfected HEK293T cells were identified by red or red/green fluorescence for whole-cell voltage clamp recordings at room temperature (21–24°C). Membrane currents were acquired using a Multiclamp 700B amplifier (Molecular Devices), filtered at 10 kHz, and sampled at 50 or 100 kHz with a Digidata 1550B interface controlled by pClamp 11 software (Molecular Devices). Series resistance was compensated by 90%, and only recordings in which the estimated residual voltage error remained below 10 mV were included in the analysis of sodium currents. Patch-clamp micropipettes were pulled from borosilicate glass capillaries (resistance 1.5–2.5 MΩ) and filled with an intracellular solution containing (in mM): 140 CsCl, 10 NaCl, 5 EGTA, 2 Mg-ATP, and 10 HEPES (pH 7.2 with CsOH). The standard bath (extracellular) solution contained (in mM): 140 NaCl, 3 KCl, 10 HEPES, 1.8 $CaCl_2$, 1.2 $MgCl_2$, and 15 glucose (pH 7.4 with NaOH). An Ag/$AgCl_2$ pellet in direct contact with the bath was used as the reference electrode. Currents were acquired without on-line leak subtraction.

Analysis of electrophysiological signals and extracted measurements was performed using Clampfit 10 (Molecular Devices), Excel (Microsoft), or Prism 10 (Graph Pad). Inactivation time constants ($\tau_i$) were obtained from fitting the decay phases of $I_{Na}$ elicited with the activation protocol to a single exponential function using Clampfit algorithms. Voltage dependence of conductance was determined from measurements of peak inward current amplitude with $G(V) = I/(V_m - V_r)$ assuming $V_r = 66$ mV, which assumes linearity in instantaneous current up to about +10 mV. GV curves with fit with a Boltzmann equation of the form

$$G(V) = G_{max} \bigg/ \left(1 + e^{\frac{zF(V-V_h)}{RT}}\right) \qquad (1)$$

where $V_h$ is the voltage at which half the maximal conductance is activated, and z reflects the voltage dependence of the channel activation equilibrium, and F/RT have their usual physical meanings. For steady-state inactivation (SSI) curves, $V_h$ reports the voltage at which half the channel population is inactivated while z reflects the voltage dependence of the distribution of channels between available and inactivated states.

For SSI curves, we utilized 500 ms prepulses to voltages from –120 to 0 mV. In our previous work with native $Na_V$ currents in rat (Martinez-Espinosa et al., 2021a) and mouse (Martinez-Espinosa et al., 2021b) chromaffin cells (CCs), we noted that 500 ms prepulses were required in order that the SSI curves reached a limiting $V_h$ value. Such currents primarily arise from $Na_V1.3$+FGF14A. In such experiments, we found that a 500 ms depolarization to 0 mV resulted in an increase in the slow recovery component that could be attributed to about 10% of the channels entering slow inactivation (Martinez-Espinosa et al., 2021a). However, other results suggest that there may be variability in both the rapidity and extent of development of slow inactivation among different $Na_V$ homologues (Vilin et al., 2012; Abdelsayed et al., 2013; Zhang et al., 2013). In our own hands, 100 ms depolarizations to 0 mV with WT $Na_V1.2$ produces minimal slow recovery, similar to another recent study on $hNa_V1.2$ (Ganguly et al., 2021). Furthermore, in Fig. 10 B, it can be seen that, for WT $Na_V1.2$ currents, a 40 Hz train of 10 pulses results in minimal development of any component of slow recovery from inactivation. Thus, the repetitive stimulation protocols used in the present work are unlikely to produce appreciable slow inactivation. However, we note that, in contrast to the results with $hNa_V1.2$, for rat $Na_V1.2$, a 500 ms depolarization to 0 mV resulted in about ~40% of the channels entering a slow inactivated condition (Abdelsayed et al., 2013). In the present experiments, it must be considered possible that the SSI curves may be impacted to some extent at the more positive prepulses by some entry into slow inactivated states. However, such an effect seems unlikely to impact on any conclusions pertinent to the effects of different FGF-A homologues on the SSI properties.

The time course of recovery from inactivation was fit with a single- or double-exponential function, including a delay term to reflect the lag typically observed before the onset of exponentiality (Capes et al., 2013), as described by the following equation:

$$I(t) = I_0 + \left[A_f\left(1 - e^{-\frac{t-t_0}{\tau_f}}\right) + A_s\left(1 - e^{-\frac{t-t_0}{\tau_s}}\right)\right] H(t - t_0) \qquad (2)$$

$$H(t - t_0) = \begin{cases} 0, t < t_0 \\ 1, t \geq t_0 \end{cases}$$

where $A_f$ and $A_s$ are the amplitudes of the fast and slow components, respectively; $\tau_f$ and $\tau_s$ are the corresponding fast and slow time constants; $t_0$ is the onset delay; $I_0$ is the value before recovery begins; and H is the Heaviside step function. Recovery parameters from $Na_V1.2$ channels expressed alone or IQM channels when expressed with FGF-A constructs were estimated using a single component of the function. Fits were performed using least-squares minimization (Excel).

## Statistics

Values reported in tables reflect means ± SD. N reports number of individual cells (biological replicates), while in parentheses, the number of individual transfections is given. Statistical comparisons were conducted separately within each $Na_V1.2$ construct (WT or IQM) among the corresponding groups (alone, +FGF14A, +FGF13A, +FGF12A, and +FGF11A) typically using one-way ANOVA, followed by Tukey's multiple comparisons test when data met the assumptions of normality and homogeneity of variance as verified by Shapiro–Wilk and Brown–Forsythe tests, respectively. When data were normally distributed but variances were unequal, Welch's ANOVA followed by Dunnett's post-hoc test was used. The statistical tests applied in each case are noted on the respective tables. To compare the effect of each FGF-A isoform between WT and IQM, an unpaired t test were performed for normally distributed data or Mann–Whitney tests when normality was not satisfied. Statistical comparisons of the slow recovery time constant ($\tau_s$) and tuning frequency ($f_{50}$) were performed assuming lognormal distributions.

Exact P values are reported in the tables. As an aid to readers, the following sets of asterisks are used on figures to denote P values of particular range: *P < 0.05; **P < 0.01; ***P < 0.001; ****P < 0.0001. P values for individual comparisons are reported with the following superscripts [11A], [12A], [13A], and [14A] to indicate specific comparison of individual FGF-A homologues to the construct given in the left column of a particular table.

## Predictions based on the LTI mechanism

A simplified version of the original formulation of a LTI mechanism as proposed by Dover et al. (2010) is given in Scheme 1.

Scheme 1.

Scheme 1 which reflects four voltage-sensor transitions leading to channel opening, and potential transitions into fast inactivated states, with $I_F$ indicative of intrinsic fast inactivation

from open states. States L2–L5 reflect the proposal that the N terminus of FGF-A homologues can independently produce a nonconducting state of Na$_V$ channels (Dover et al., 2010). To what extent closed-state inactivation can arise from either of the two inactivation pathways remains incompletely understood.

When Na$_V$ channels are driven rapidly to high open probabilities, inactivation from closed states is likely reduced, such that overall inactivation behavior can be largely approximated by inactivation from open states, allowing the simplification given in Scheme 2. Entry and exit from I$_F$ occurs via rates $k_i$ and $k_{-i}$, while entry and recovery with I$_{LTI}$ are governed by $k_{lti}$ and $k_{-lti}$.

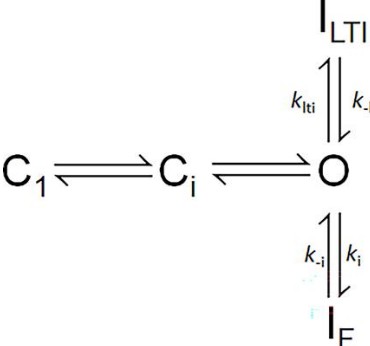

Scheme 2.

For most of the protocols employed here, we utilize strong depolarizations (≥0 mV) intended to produce rapid and near maximal channel activation, such that inactivation can be effectively modelled via transitions from the open state (Scheme 2). Scheme 2 makes a number of straightforward predictions that can be applied to compare inactivation mediated by different FGF-A isoforms.

### Onset of inactivation
Whereas the time constant of inactivation of Na$_V$ channels that inactivate only by intrinsic fast inactivation at strong depolarizations will reflect: $\tau_i \sim 1/k_i$, the macroscopic time constant of inactivation onset at strong activation conditions for channels arising from Na$_V$+FGF-A will approximated by

$$\tau_i \sim 1/(k_{lti} + k_i) \tag{3}$$

Thus, for a complex of Na$_V$1.2+FGF-A and assuming independence of the two inactivation processes, the macroscopic time constant of the observed inactivation would be expected to be faster than that for Na$_V$ channels lacking the FGF-A isoform. To our knowledge, the relative contributions of $k_{lti}$ and $k_i$ to macroscopic fast inactivation of Na$_V$+FGF-A complexes have not been evaluated.

### Recovery from inactivation
Scheme 2 predicts two independent components of recovery from inactivation, from which information about the underlying molecular transitions might be inferred. Each recovery time constant is predicted to be independent of the fraction of channels that occupy I$_F$ or I$_{LTI}$ at the time of repolarization as noted in previous work (Martinez-Espinosa et al., 2021a; Martinez-Espinosa et al., 2021b). Furthermore, if return to the open state (O)

from both inactivated states, I$_{LTI}$ and I$_F$, only occurs rarely at very strong activation potentials, then following repolarization the ratio of channels that recover from I$_{LTI}$ vs. that from I$_F$ will be defined by the fraction of channels recovering via slow recovery, as follows

$$A_s = k_{lti}/(k_{lti} + k_i) \tag{4}$$

For channels in which the fast and slow components are of identical amplitude, this would indicate that the intrinsic rates of inactivation onset, $k_{lti}$ and $k_i$, are identical. Given the apparent lack of interconversion between I$_{LTI}$ and I$_F$ once inactivation has occurred (Martinez-Espinosa et al., 2021a; Martinez-Espinosa et al., 2021b), the slow and fast time constants of recovery from inactivation then explicitly yield rate constants for recovery from I$_{LTI}$ and I$_F$ as defined by

$$\tau_s = 1/k_{-lti} \tag{5}$$

and

$$\tau_f = 1/k_{-i} \tag{6}$$

Previous studies of LTI mechanism have not evaluated to what extent there are differences among different FGF-A isoforms.

### Use of an inactivation-removed Na$_V$
Further support regarding the role of FGF-A homologues in inactivation has been gained from coexpression of FGF-A homologues with inactivation-removed Na$_V$ isoforms. In one such case (Dover et al., 2010), when FGF13A was coexpressed with a non-inactivating Na$_V$1.6 subunit (F1478Q), the resulting currents exhibited rapid and relatively complete inactivation, although quantitative evaluation of relative rates of entry into I$_F$ and I$_{LTI}$ was not undertaken. In the absence of I$_F$, a single time constant of recovery from inactivation is expected, which should correspond closely to the time constant of slow recovery for Na$_V$+FGF-A currents. Similarly, the rate of onset of inactivation mediated by FGF-A homologues on Na$_V$1.2-IR provides an independent measure of $k_{lti}$. These expectations have not been explored in previous work.

Overall, the general model encapsulated in Schemes 1 and 2 has provided a reasonable description not only of the inactivation behavior of FGF-A–associated channels but also use-dependent changes in Na$_V$ availability during repetitive stimulation (Dover et al., 2010; Milescu et al., 2010; Goldfarb, 2012; Goldfarb, 2024; Navarro et al., 2020; Martinez-Espinosa et al., 2021a; Martinez-Espinosa et al., 2021b).

### Calculations of use-dependent reductions in Na$_V$ current
For trains of 10 10-ms depolarizations (P1..P10), we assume that all Na$_V$ channels are available for activation at the onset of P1, that during each depolarization all available channels become inactivated by the end of the depolarization, and that relative fractional entry of available channels into either I$_F$ or I$_{LTI}$ is defined by f(I$_F$) = $k_i/(k_i + k_{lti})$ and f(I$_{LTI}$) = $k_{lti}/(k_i + k_{lti})$. Between each depolarization, a given recovery interval dependent on train frequency permits calculation of the fraction of I$_F$ and I$_{LTI}$ inactivated channels that return to availability (resting state, C)

based on the respective $k_{-i}$ and $k_{-lti}$ recovery rates with some channels also remaining in either $I_F$ and $I_{LTI}$. Through a set of 10 depolarizations and the 9 intervening recovery intervals, the fraction of available channels preceding each of the i = 1..10 depolarizations (fC(i)) and the fractional occupancy in $I_F$ and $I_{LTI}$ ($A_f(i)$ and $A_s(i)$) after each depolarization can be sequentially determined. fC(i) is taken to report on pulse-dependent changes in $Na_V$ availability during a given train.

### Online supplemental material

Fig. S1 evaluates use-dependent diminution of $Na_V1.2\_IQM$ with various FGF-A homologues. Fig. S2 compares the use-dependent diminution of $Na_V1.2\_IQM$ currents at different train frequencies and compares it with predictions of the allosteric model of LTI. Table S1 provides information on fits of a Boltzmann function to GV curves for $Na_V1.2$ without and with FGF-A homologues. Table S2 provides parameters of fits of Boltzmann functions to SSI curves. Table S3 provides parameters for fits of Boltzmann functions to GV curves for $Na_V1.2\_IQM$ without and with FGF-A homologues. Table S4 provides parameters for fits of Boltzmann functional to SSI curves for $Na_V1.2\_IQM$ without and with FGF-A homologues. Table S5 compares Boltzmann fit parameters with GV curves of WT and IQM without and with FGF-A homologues, Boltzmann fit parameters to SSI curves of WT and IQM without and with FGF-A homologues, and compares time constants of onset of inactivation and slow recovery from inactivation for WT and IQM without and with FGF-A homologues. Table S6 summarizes use-dependent changes in FGF-A–mediated recovery from LTI.

## Results

### Properties of currents arising from heterologous expression of FGF-A homologues with $Na_V1.2$

$Na_V1.2$ channels were expressed in HEK293T cells alone (WT) and then together with each of the four FGF-A homologues, FGF14A, 13A, 12A, and 11A (Fig. 1 A). Overall, the current families are quite similar, although the development of inactivation appears slower when $Na_V1.2$ is coexpressed with any of the FGF-A isoforms. The normalized GV curves (Fig. 1 B) in the presence of FGF-A homologues are not obviously different from WT, yielding similar estimates for $V_h$ (Fig. 1 C) and voltage-dependence (z; Fig. 1 D). Given the inactivation-mediated reduction of peak current, the GV curves only roughly approximate the true voltage dependence of fractional activation but are a useful measure for qualitative comparisons. Here, any effect of FGF-A homologues on apparent GV relationships was minimal (statistical comparisons are given in Table S1).

From traces such as those in Fig. 1 A, we fit the decay phase of currents activated by voltage steps positive to –30 mV with a single exponential function to obtain estimates of inactivation time constants (Fig. 1 E). For each of the FGF-A homologues, there is a suggestion of a somewhat slowed onset of inactivation (Fig. 1 F and Table S1) in comparison with $Na_V1.2$ expressed alone, which becomes somewhat clearer at more positive voltages. As noted in the Materials and methods, if intrinsic fast inactivation ($k_i$) is a competitive process with FGF-A–mediated

inactivation ($k_{lti}$), one would expect that inactivation with both inactivation mechanisms intact would be faster than for intrinsic fast inactivation alone. This is clearly not observed. This issue will be discussed in more detail below.

SSI curves for each FGF-A homologue expressed with $Na_V1.2$ were generated with a 500 ms prepulse over voltages from –120 to 0 mV (Fig. 2 A), as described in the Materials and methods. 500 ms insured that the SSI curves reflect the equilibrium of both intrinsic fast inactivation and LTI (Martinez-Espinosa et al., 2021a). However, it leaves open the possibility that some entry into slow inactivation may impact on the shape of SSI curves at the more positive voltages. Here, the resulting SSI curves revealed that each FGF-A homologue produces a rightward shift of about 10–20 mV (Fig. 2, B–E; and Table S2), similar to previous observations reported for $Na_V1.6$ coexpression with 12A, 13A, and 14A (Dover et al., 2010). Although some FGF-B homologues can also shift SSI inactivation rightward (Dover et al., 2010; Goldfarb, 2024), this has not been universally noted for all FGF-B subunits, perhaps dependent on the identity of the associated $Na_V$ channel (Goldfarb, 2024). It should be kept in mind that the parallel rightward shifts in SSI curves observed here with the FGF-A isoforms support the idea that, when $Na_V1.2$ is coexpressed with each of these FGF-A homologues, all channels in a cell are assembled with the appropriate FGF-A homologue. We also noted a modest increase in the voltage dependence of the SSI relationship when $Na_V1.2$ is coexpressed, particularly with FGF13A and FGF14A (Fig. 2, B, C, and G). This might be taken to reflect a different voltage dependence of transitions involved in LTI compared with intrinsic fast inactivation, although that issue is not being addressed here.

### Fractional entry into and rates of recovery from FGF-A–mediated LTI differ among FGF-A homologues when coexpressed with $Na_V1.2$

The distinguishing feature of LTI current is that it leads to two distinct components of recovery from inactivation (Martinez-Espinosa et al., 2021a). Whether recovery from inactivation is monitored after a single depolarizing step or a train of depolarizations that drives a larger fraction of channels into LTI, two time constants of recovery are observed, one corresponding approximately to recovery from $I_F$ and one from $I_{LTI}$. In the protocol used here, a pair of depolarizing steps to 0 mV ($P_1$ and $P_2$, Fig. 3 A, top) were separated by a recovery interval at –80 mV varying from 0.1 to 15 s. Prior to the $P_1$ step, each cell was held at –80 mV for 500 ms. Recovery from inactivation at –80 mV was determined for WT $Na_V1.2$ expressed alone or with each of the four FGF-A homologues (Fig. 3 A). For $Na_V1.2$ alone, inspection of the traces qualitatively indicates that recovery is complete within 100 ms, whereas for coexpression with the different FGF-A homologues a substantial fraction of current remains inactivated at 100 ms. Fractional recovery for each construct was determined by plotting the recovery in amplitude during the current evoked by the $P_2$ depolarization relative to the response during the $P_1$ depolarization (Fig. 3 B). The recovery time course for $Na_V1.2$ currents expressed alone was best described by a single exponential time course with $\tau_f$ = 8.7 ± 2.3 ms (Fig. 3 B). In contrast, when $Na_V1.2$ is coexpressed with FGF-A homologues, a double exponential function is required

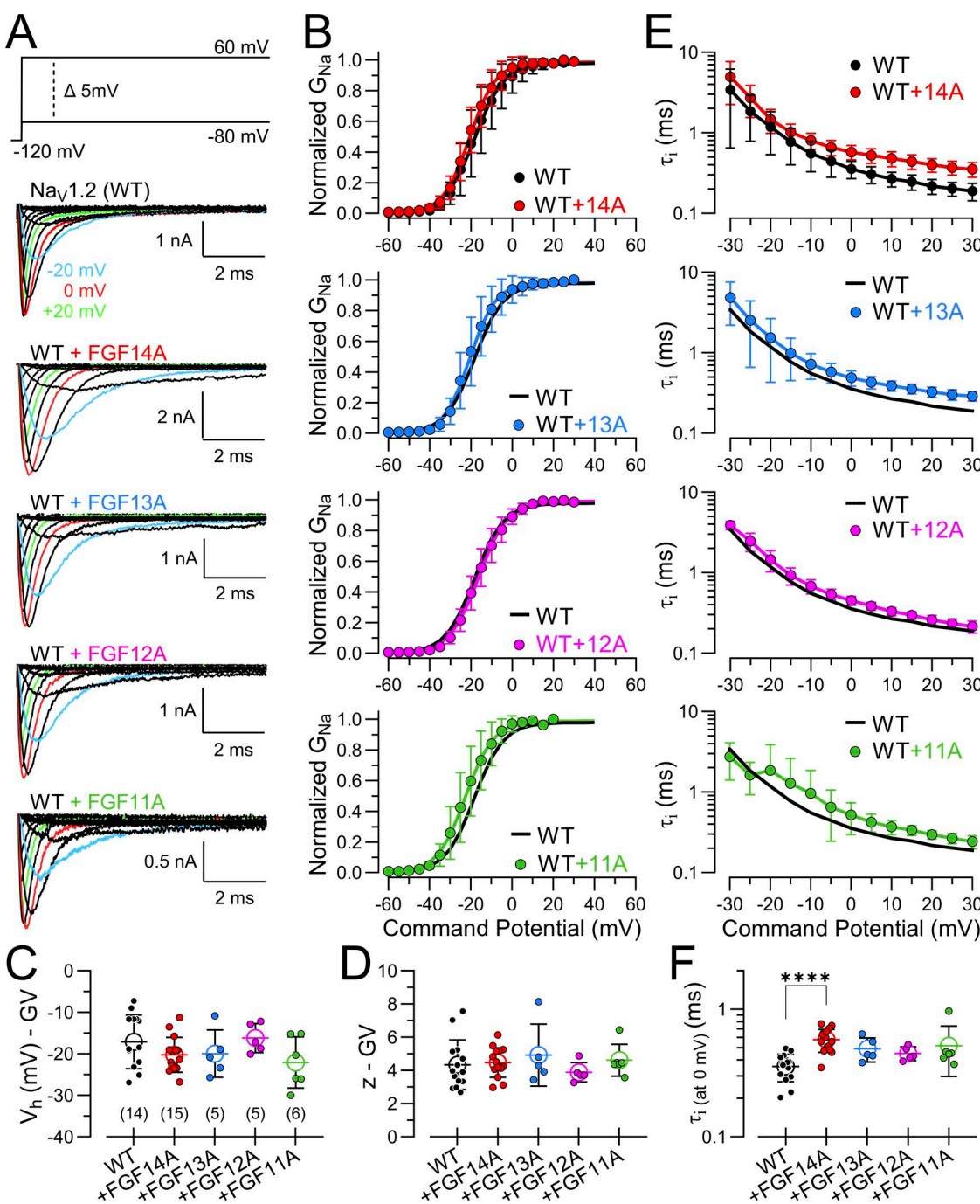

Figure 1. **Properties of Na$_V$1.2 currents coexpressed with different FGF-A homologues. (A)** Example Na$_V$1.2 currents recorded from HEK293T cells following transient expression of Na$_V$1.2 cDNA alone (WT) or together with cDNA for the indicated A-type FGF proteins (FGF14A, FGF13A, FGF12A, and FGF11A). Currents were evoked by 25-ms depolarizing pulses from –80 to +60 mV in 5 mV increments (protocol on top). **(B)** Effect of FGF-A-homologues on voltage-dependent activation of Na$_V$1.2 channels. Averages of individual GV relationships were generated as described in Materials and methods. Solid lines represent Boltzmann fits to the data. Values in Table S1. **(C and D)** Plots of mean $V_h$ (C) and z (D) parameters from averaged GV curves with points plotting values from individual cells. **(E)** Voltage dependence of fast inactivation time constants ($\tau_i$) for WT Na$_V$1.2 is compared with those resulting from coexpression with each of the FGF-A isoforms. $\tau_i$ was obtained by fitting the decay phase of each current trace in panel A with a single exponential function (see Materials and methods). (Table S1 for statistical comparisons). **(F)** Comparison of $\tau_i$ at 0 mV for each condition (mean ± SD) with statistics given in Table S1. Statistical comparisons in C and D were performed using an ANOVA test with Tukey correction for multiple comparisons. In F, comparison was made with Welch ANOVA with Dunnett's correction for multiple comparisons. ****$P < 0.0001$.

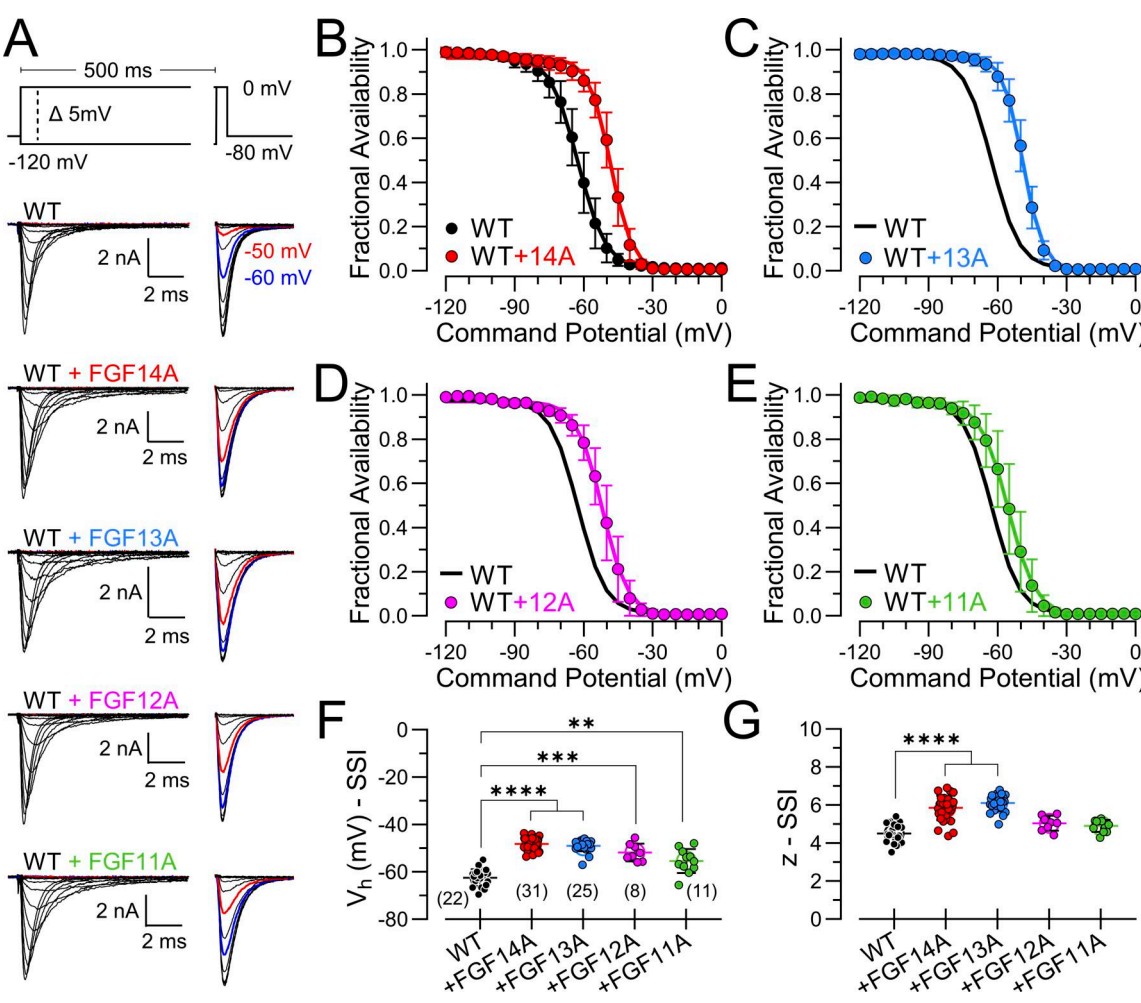

Figure 2. **All FGF-A homologues produce a similar rightward/depolarizing shift in SSI. (A)** Families of currents resulting from the indicated SSI protocol (top) for $Na_V1.2$ alone (top traces) or when coexpressed with FGF14A (red), FGF13A (blue), FGF12A (magenta), or FGF11A (green). Inactivation was elicited by a 500 ms conditioning pulse (left traces) to potentials over the range of –120 to 0 mV in 5 mV increments prior to a test pulse at 0 mV (right traces). **(B)** Effect of FGF14A on voltage dependence of SSI of $Na_V1.2$ channels. Individual fractional availability was determined from $I_{Na}$ evoked at 0 mV (test pulse) following a given conditioning potential with normalization to the maximum $I_{Na}$. Solid lines show Boltzmann fits to averaged data points (mean ± SD). **(C)** SSI curves and fits for $Na_V1.2$+FGF13A. **(D)** $Na_V1.2$+FGF12A. **(E)** $Na_V1.2$+FGF11A. **(F and G)** Mean, SD, and individual estimates of $V_h$ (F) and $z$ (G) estimated from SSI curves (B–E) for $Na_V1.2$ without and with the different FGF-A homologues. Numbers along bottom are the number of cells in each case. Fit parameters, $N$, and P values are summarized in Table S3. **P < 0.01, ***P < 0.001, and ****P < 0.0001.

to describe the recovery time course (Fig. 3 B). With WT $Na_V1.2$+FGF-A coexpression, the fast time constants ($\tau_f$) of recovery defined by the double exponential fits are appreciably faster than for $Na_V1.2$ alone (Fig. 3 C, top; Table 2). Among different FGF-A homologues, both the time constants of slow recovery ($\tau_s$; Fig. 3 C, bottom) and the fractional amplitudes of the slow recovery component ($A_s$, Fig. 3 D, bottom) vary markedly among FGF-A isoforms. The time constant of slow recovery, reflecting the stability of LTI among FGF-A homologues, varies in the order FGF13A > FGF14A~FGF12A > FGF11A (Fig. 3 C, bottom). In contrast, the amplitude of the slow component, $A_s$, decreases in the order FGF14A > FGF13A~FGF11A > FGF12A (Fig. 3 D). This indicates that recovery from inactivation mediated by each FGF-A homologue occurs with a distinct, characteristic rate (given by Eq. 3). Furthermore, the differences in the amplitudes of the fraction of channels that recover via slow recovery reveal that there are also differences among FGF-A

homologues in the rates of development of their inhibitory effects (Eq. 2). In short, there are both differences among FGF-A isoforms in the forward rates of FGF-A–mediated fast inactivation and the recovery rate among different FGF-A homologues, despite the general homologies (Table 1) among the FGF-A N termini.

### FGF-A homologues produce rapid inactivation of fast inactivation-removed $Na_V1.2$ channels

To evaluate the kinetic properties of FGF-A–mediated inactivation in the absence of intrinsic fast inactivation, we expressed the $Na_V1.2\_IQM$ (IQM) construct alone and then together with each of the FGF-A subunits. Over the duration of a 25 ms voltage step, the IQM currents exhibit little inactivation (Fig. 4 A). However, when IQM is coexpressed with any of the four FGF-A isoforms, the currents exhibit rapid and largely complete inactivation. The resulting GVs (Fig. 4 B) for IQM without and with coexpression with each of the FGF-A

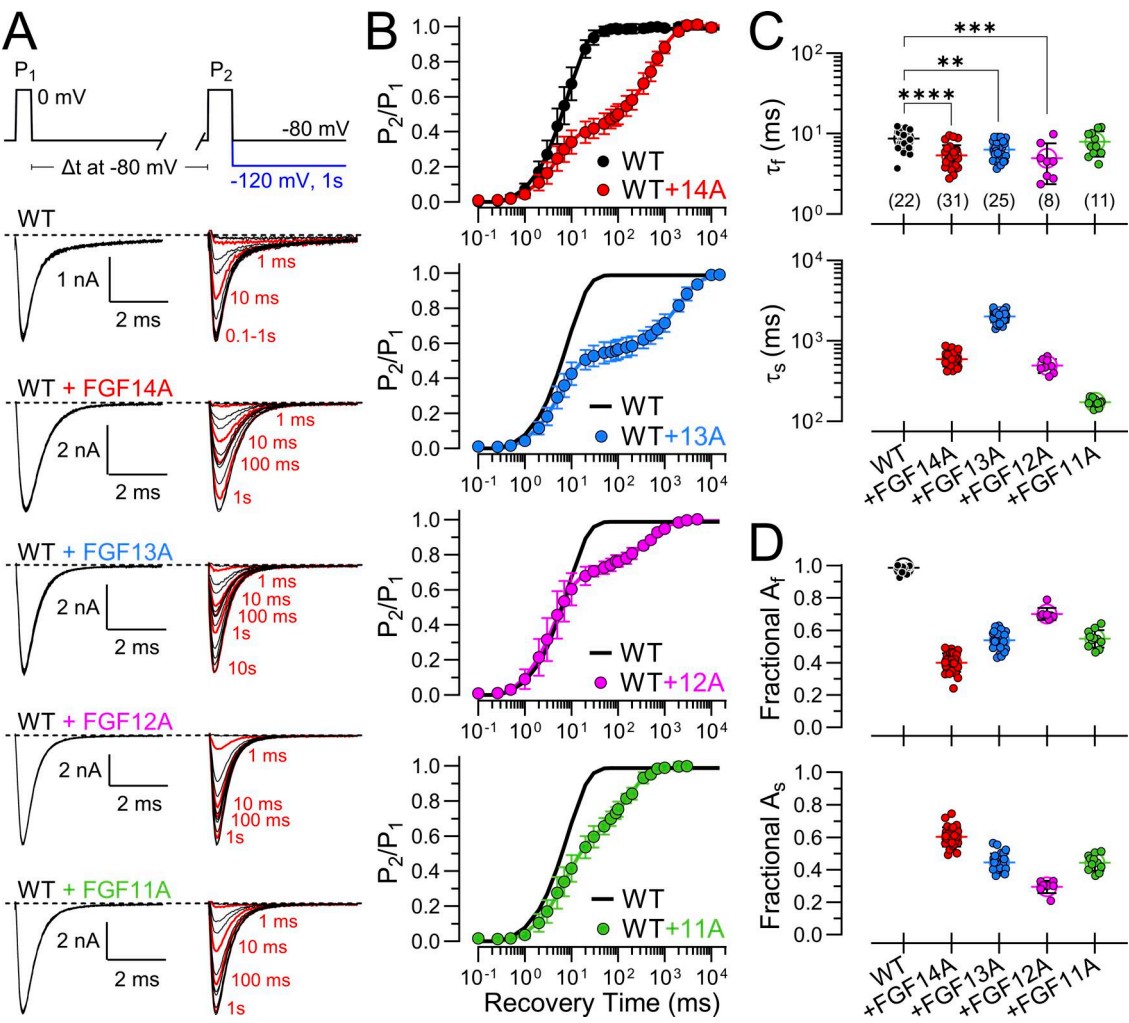

**Figure 3. Onset and recovery from LTI differs among FGF-A homologue. (A)** The indicated paired pulse protocol (top) was used to assess the time course of recovery from inactivation. Cells were held at –80 mV. An initial 10 ms depolarization to 0 mV (pulse 1, $P_1$) was followed after variable recovery times (0.1 ms to 15 s) by a 25 ms step to 0 mV (pulse 2, $P_2$). A 1 s step to –120 mV separated each paired pulse pair to permit full recovery from inactivation. From top to bottom, superimposed $P_1$- (left) and $P_2$- (right) elicited currents for $Na_V1.2$ alone (WT) or coexpressed with each FGF-A isoform (FGF14A, FGF13A, FGF12A, and FGF11A) are shown. Red traces show $I_{Na}$ following recovery intervals of 1, 10, 100, 1,000 and 10,000 ms. **(B)** Averaged time course of fractional recovery following inactivation for WT (black), WT+FGF14A (red), WT+FGF13A (blue), WT+FGF12A (magenta), or WT+FGF11A (green). $I_{Na}$ elicited by $P_2$ was normalized to $I_{PEAK}$ measured in $P_1$ ($P_2/P_1$) and plotted as a function of different recovery intervals. Solid lines show single (for WT) or double (for WT+FGF-A) exponential fits to data points (mean ± SD) (values in Table 4). **(C)** Scatter plots of fast ($\tau_f$, top panel) and slow ($\tau_s$, bottom panel) time constants of recovery for each construct with means ± SD along with values for individual cells. See Table 2 for statistical comparisons. **(D)** Scatter plots of fractional amplitudes of fast ($A_f$: top panel) and slow ($A_s$: bottom panel) recovery components for each construct along with values for individual cells. Data are expressed as mean ± SD (Table 2). Although $A_f \sim 1 - A_s$, each parameter is plotted separately here for illustrative purposes. Fit parameters, statistics, and $N$ are provided in Table 2. **P < 0.01, ***P < 0.001, and ****P < 0.0001.

homologues support the view that each FGF-A isoform has minimal effect on the voltage dependence of WT $Na_V1.2$ activation (Fig. 4, C and D; and Table S3). Time constants of inactivation mediated by the FGF-A homologues when coexpressed with $Na_V1.2\_IQM$ (Fig. 4 E) were slower than either those for WT $Na_V1.2$ expressed alone or when coexpressed with the FGF-A homologues. Representative examples of inactivation of IQM+FGF-A currents (Fig. 4 F) at 0 mV highlight the much slower inactivation onset when mediated only by the FGF-A LTI process, in comparison with WT or WT+FGF-A's. Although time constants of inactivation measured at 0 mV were fairly similar among the four FGF-A homologues

(Fig. 4 G), the rank order of inactivation onset was FGF11A~ FGF14A > FGF12A >FGF13A (also Table 3).

For each of the IQM+FGF-A currents, we also determined fractional availability curves for FGF-A–mediated inactivation of IQM (Fig. 5 A). Similar to effects of A-homologues on SSI of WT $Na_V1.2$, the fractional availability for each IQM+FGF-A complex exhibited a rightward 10–20 mV shift compared with inactivation for native $Na_V1.2$ (Fig. 5 B and Table S4).

Expression of $Na_V1.2\_IQM$ with each FGF-A homologue also permitted direct examination of the rate of recovery from LTI when the intrinsic fast inactivation process is absent. Fig. 6 A compares recovery from inactivation using the standard paired

**Table 2.  Properties of recovery from inactivation for $Na_V1.2$ + FGF-A isoforms**

| Construct | Recovery from inactivation at −80 mV | | | | | | | | |
|---|---|---|---|---|---|---|---|---|---|
| | $\tau_f$ (ms) | P value | $A_f$ | P value | $\tau_s$ (ms) | P value | $A_s$ | P value | N |
| $Na_V1.2$ | 8.7 ± 2.3 | <0.0001 [14A]<br>0.002 [13A]<br>0.0004 [12A]<br>0.87 [11A] | 0.99 ± 0.02 | <0.0001 [14A]<br><0.0001 [13A]<br><0.0001 [12A]<br><0.0001 [11A] | | | | | 22 (16) |
| +FGF14A | 5.4 ± 1.8 | 0.46 [13A]<br>0.99 [12A]<br>0.007 [11A] | 0.40 ± 0.06 | <0.0001 [13A]<br>0.05 [12A]<br><0.0001 [11A] | 592.9 ± 111.5 | <0.0001 [13A]<br>0.05 [12A]<br><0.0001 [11A] | 0.60 ± 0.06 | <0.0001 [13A]<br><0.0001 [12A]<br><0.0001 [11A] | 31 (18) |
| +FGF13A | 6.3 ± 1.7 | 0.50 [12A]<br>0.21 [11A] | 0.54 ± 0.06 | <0.0001 [12A]<br>0.99 [11A] | 2017.6 ± 324.0 | <0.0001 [12A]<br>0.99 [11A] | 0.45 ± 0.06 | <0.0001 [12A]<br>0.99 [11A] | 25 (16) |
| +FGF12A | 5.0 ± 2.6 | 0.02 [11A] | 0.70 ± 0.04 | <0.0001 [11A] | 497.2 ± 97.7 | <0.0001 [11A] | 0.29 ± 0.04 | <0.0001 [11A] | 8 (2) |
| +FGF11A | 7.9 ± 2.7 | | 0.55 ± 0.05 | | 173.1 ± 19.7 | | 0.44 ± 0.05 | | 11 (7) |

In all tables, parameter values are means ± SD, while N = number of biological replicates (cells) with number of transfections given in parentheses. Statistical analyses were performed using one-way ANOVA, followed by Tukey's multiple comparisons test (for $\tau_{slow}$ parameter, a lognormal distribution was used to satisfy ANOVA assumptions of normality and homogeneity of variance). In all tables, [14A], [12A], [13A], and [11A] represent paired comparisons between WT+FGF14A, WT+FGF12A, WT+FGF13A, or WT+FGF11A, respectively, with the corresponding construct in the first column.

pulse protocol for WT $Na_V1.2$, WT+FGF14A, and then IQM+FGF14A. Recovery from inactivation mediated by any of the FGF-A homologues when expressed with IQM was described exclusively by a single exponential time course (Fig. 6, B–E), providing a clear definition of the rate of recovery from inactivation mediated by each homologue. In each example, the averaged recoveries for IQM with each FGF-A isoform are compared with that of the averaged WT $Na_V1.2$ and the corresponding averaged WT $Na_V1.2$+FGF-A homologue.

Fig. 7 summarizes key features of the A-isoform–mediated effects on $Na_V1.2$ and $Na_V1.2$_IQM inactivation behavior. In the absence of normal $I_F$, each FGF-A isoform independently mediates fast inactivation, but at rates slower than that of intact $I_F$ alone (Fig. 7 A). Among the different FGF-A isoforms, the onset of the isolated LTI differs among FGF-A isoforms with the rank order given above, although the differences appear rather minor. In contrast, for the IQM+FGF-A currents, differences in recovery from LTI are clearly distinct from each other and essentially unchanged from those with fast inactivation intact (Fig. 7 B). For recovery, the rank order of the recovery rate is FGF11A > FGF12A∼FGF14A > FGF13A. Direct measures of effects of each FGF-A homologue on WT $Na_V1.2$ vs. $Na_V1.2$_IQM for activation behavior, SSI behavior, and time constants of recovery from inactivation are provided in Table S5, respectively.

**Probing determinants of inactivation onset, recovery, and SSI with mutant FGF14A N termini**

The above analysis indicates that, despite the extensive N-terminal similarity among FGF-A homologues, there are major differences in stability of the LTI-inactivated state once inactivation has developed. That the rate of recovery of any given N terminus can be quantitatively assessed via the paired pulse protocol potentially provides a powerful tool for examination of the determinants of FGF-A interactions with a given $Na_V$ channel.

Mutational evaluation of residues in the N terminus of the FGF13A subunit, along with experiments with isolated FGF13A N-terminal peptides (2–21), have supported the idea that the FGF-A–mediated LTI arises from a specific N-terminal inactivation segment (Dover et al., 2010; Venkatesan et al., 2014). Furthermore, such results have led to the proposal that the rate of onset of LTI likely reflects the association of the N-terminal inactivation domain with a position that inhibits ion flux, while recovery from inactivation reflects dissociation of the blocking particle from its site of occupancy (Dover et al., 2010). Whether such a simple occlusion model applies to LTI has not been fully established, but nothing as yet excludes that possibility. Within this context, it is convenient to consider that the forward rate of inactivation reflects a molecular association process, while recovery may reflect dissociation. Here, we take advantage of the paired pulse procedure to probe whether there may be determinants that may differentially impact on forward vs. reverse rates of LTI. This evaluation is not meant to be a systematic evaluation of determinants of forward and reverse rates, but simply a test of a limited set of mutations in the FGF14A N terminus (Fig. 8 A) to confirm the utility of this approach. Two constructs (LI/AA and 5Q) recapitulate those previously studied

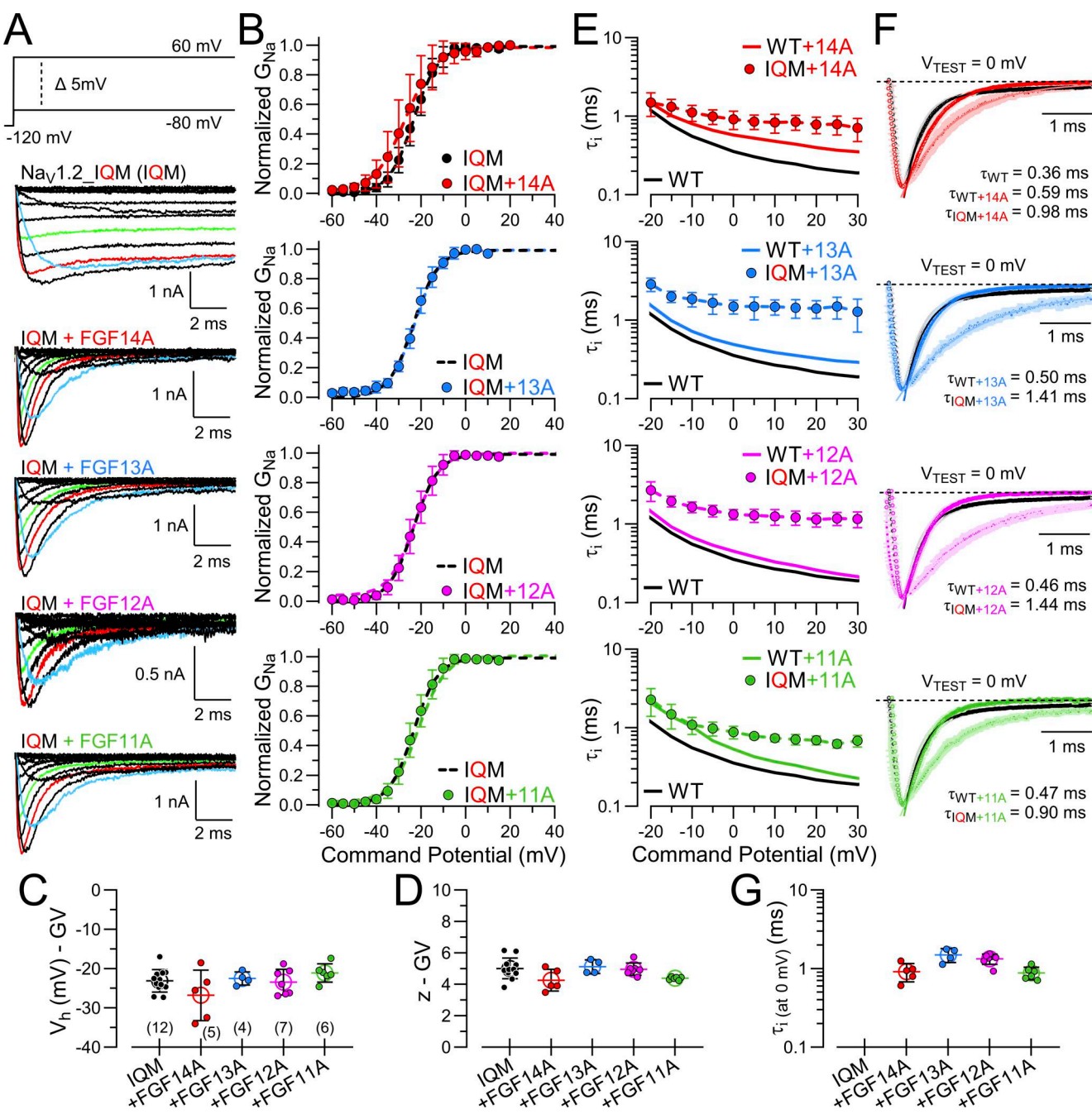

Figure 4. **FGF-A homologues directly produce fast inactivation of inactivation-removed Na$_V$1.2 channels. (A)** Example Na$_V$ currents arising from Na$_V$1.2_ IQM channels transiently expressed alone (IQM) or coexpressed with the indicated FGF-A subunits (FGF14A, FGF13A, FGF12A, and FGF11A). Currents were evoked by the indicated protocol (top). Colored traces highlight current at −20 mV (blue), 0 mV (red), and +20 mV (green). **(B)** GV curves for each construct (IQM (black), IQM+FGF14A (red), IQM+FGF13A (blue), IQM+FGF12A (magenta), and IQM+FGF11A (green) are shown with peak current evoked by each test pulse normalized to I$_{MAX}$. Dashed lines are Boltzmann fits to data points (mean ± SD). **(C and D)** Graphs of V$_h$ (C) and z (D) from GV curves for each construct along with values for individual cells. Data are expressed as mean ± SD with statistics given in Table S3. **(E and F)** Voltage dependence of mean inactivation onset for IQM+FGFA homologues with solid block line for Na$_V$1.2 alone, solid colored line for Na$_V$1.2+FGF-A, and symbols for IQM+FGF-A. Red, FGF14A; blue, FGF13A; magenta, FGF12A; green, FGF11A. **(F)** Comparisons of inactivation time course following activation at 0 mV with WT Na$_V$1.2 in black, Na$_V$1.2+FGF-A in a colored line, and IQM+FGF-A as symbols with shaded dotted lines. **(G)** Mean inactivation time constants ± SD at 0 mV for each FGF-A homologue expressed with Na$_V$1.2_IQM, along with values from individual cells.

for FGF13A (Dover et al., 2010). In both cases, a 10-pulse train at 5 Hz produced no reduction in peak I$_{Na}$ (Fig. 8 B) consistent with the impact of those mutations in FGF13A. In contrast, in a construct in which only two of the residues in the 5Q construct were mutated (2Q: R17Q/W21Q), we observed a partial reduction in the use-dependent diminution of peak I$_{Na}$, which was about half of that produced by WT FGF14A at the 5 Hz train frequency (Fig. 8, B and D). For each construct, we applied the standard

**Table 3.  Time constants of inactivation onset and recovery for Na_V1.2_IQM expressed with FGF A-isoforms**

| Constructs | Inactivation τ (0 mV) | | | Slow recovery from inactivation (–80 mV) | | | |
|---|---|---|---|---|---|---|---|
| | $\tau_i$ (ms) | P value | N | $\tau_s$ (ms) | P value | $A_s$ | N |
| Na_V1.2 | 0.36 ± 0.09 (Table S1) | | 14 | | | | |
| Na_V1.2_IQM | | | | | | | |
| +FGF14A | 0.92 ± 0.24 | 0.005 [13A]<br>0.02 [12A]<br>0.99 [11A] | 5 (5) | 565.5 ± 88.3 | <0.0001 [13A]<br>>0.99 [12A]<br><0.0001 [11A] | 0.98 ± 0.03 | 4 (4) |
| +FGF13A | 1.50 ± 0.30 | 0.64 [12A]<br>0.002 [11A] | 4 (3) | 1831.5 ± 383.5 | <0.0001 [12A]<br><0.0001 [11A] | 0.95 ± 0.03 | 6 (4) |
| +FGF12A | 1.33 ± 0.20 | 0.008 [11A] | 7 (5) | 565.4 ± 112.5 | <0.0001 [11A] | 1.00 ± 0.03 | 10 (6) |
| +FGF11A | 0.88 ± 0.16 | | 6 (4) | 155.8 ± 34.5 | | 0.97 ± 0.02 | 10 (4) |

Statistical analyses were performed using one-way ANOVA, followed by Tukey's multiple comparisons test (for $\tau_{slow}$ parameter, a lognormal distribution was used to satisfy ANOVA assumptions of normality and homogeneity of variance). Rank order of derived rate constants.
$k_{lti}$: FGF11A ~ FGF14A > FGF12A > FGF13A.
$k_{-lti}$: FGF11A > FGF12A ~ FGF14A > FGF13A.

paired-pulse protocol (Fig. 8 C) to determine the impact of a given mutation on the fast and slow time constants (if present) of recovery from inactivation and also the fractional entry ($A_s$) into the LTI condition. For the LI/AA and 5Q mutations, there was no

slow recovery to be measured, which was also true of FGF14A_ ΔNT and FGF14B constructs (Fig. 8, D–F). However, for 2Q, slow recovery from inactivation was readily measured (Fig. 8 F and Table 4). Intriguingly, the amplitude of the slow recovery

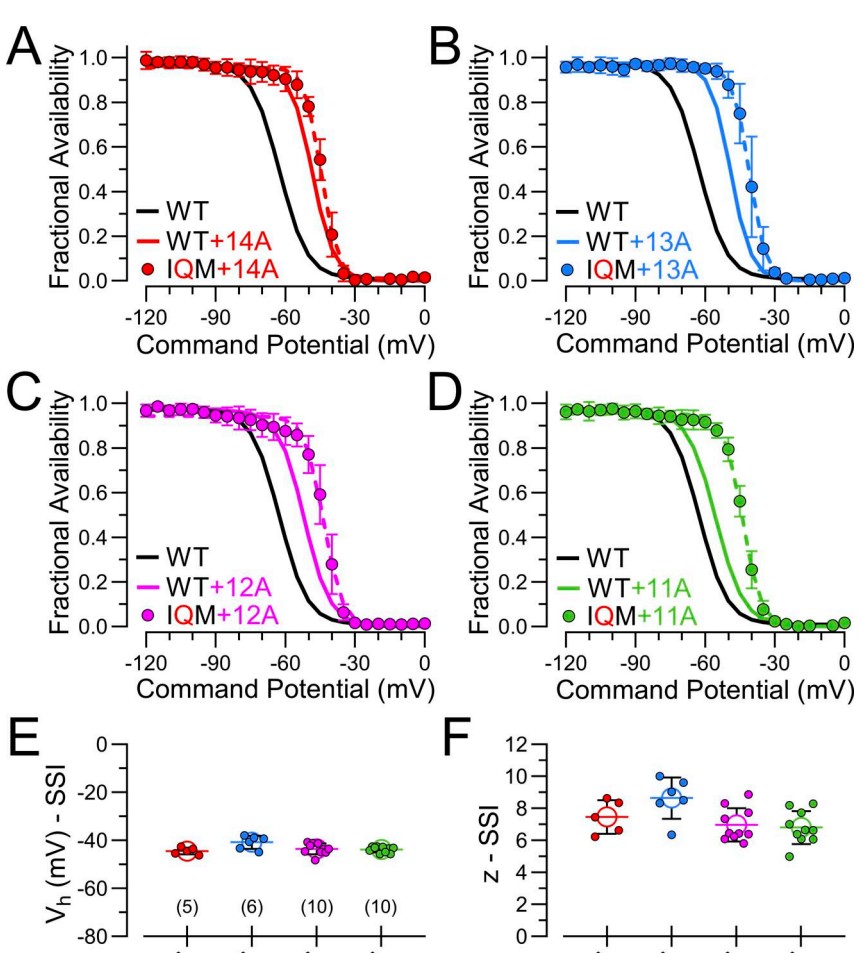

**Figure 5.  SSI behavior for Na_V1.2_IQM expressed with different FGF-A homologues. (A)** SSI protocols (identical to those in Fig. 2) were used to examine fractional availability as a function of voltage. Solid lines correspond to WT Na_V1.2 alone (black) and Na_V1.2+FGF14A (red), while symbols show values for FGF14A expressed with Na_V1.2_IQM (solid symbols). Dashed line is fitted Boltzmann function for the IQM+FGF14A points. **(B–D)** Similar comparisons for FGF13A (B), FGF12A (C), and FGF11A (D). **(E)** Mean ± SD and individual values for $V_h$ estimates from Boltzmann fits from A–D with the indicated number of cells in each case given below. **(F)** Comparison of effective valence, z, of SSI for each Na_V1.2_IQM+FGF-A homologue. Statistical comparisons are in Table S4.

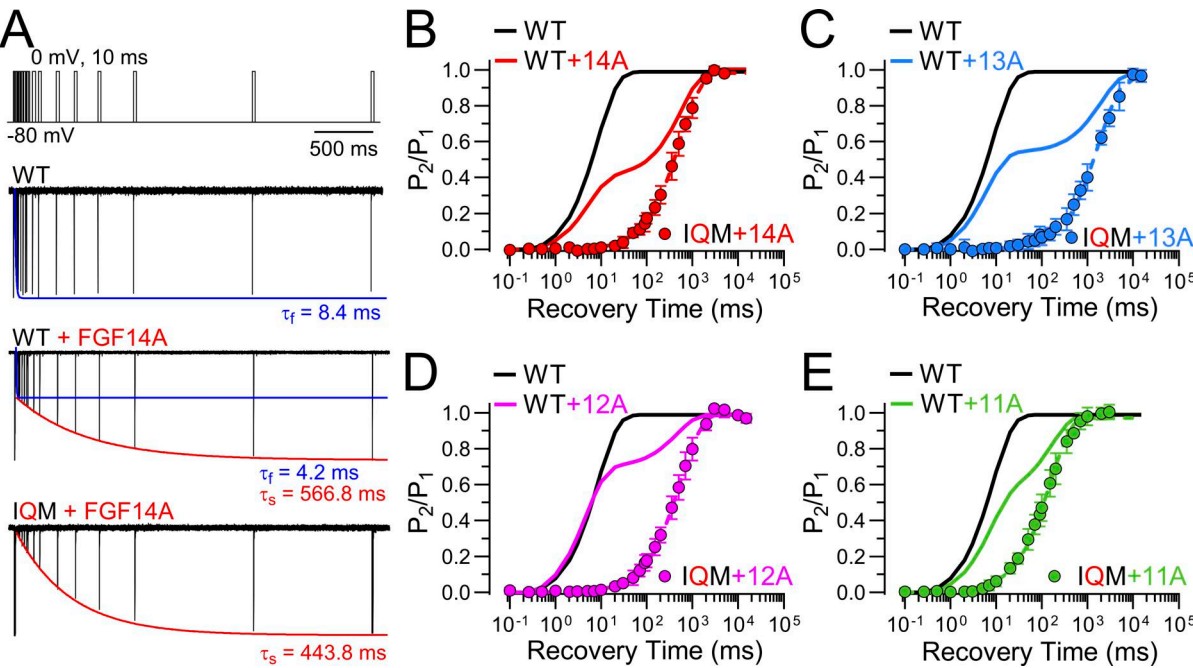

Figure 6. **Rate of recovery from inactivation of Na$_V$1.2_IQM varies among different FGF-A homologues. (A)** The paired pulse protocol on the top was used to examine the time course of recovery from inactivation for WT Na$_V$1.2, Na$_V$1.2+FGF14A, and then Na$_V$1.2_IQM+FGF14A, as indicated. For Na$_V$1.2 alone, only a single fast component of recovery is observed. For Na$_V$1.2+FGF14A, two components of recovery can be seen with slow recovery contributing a bit more than half. For Na$_V$1.2_IQM+FGF14A, only a single slow component of recovery is observed. **(B)** Averaged recovery from inactivation for Na$_V$1.2_IQM+FGF14A is plotted along with results shown earlier for Na$_V$1.2 alone (black line) and Na$_V$1.2+FGF14A (solid colored line). **(C–E)** Similar comparisons for recovery for Na$_V$1.2_IQM+FGF13A (C), Na$_V$1.2_IQM+FGF12A (D), and Na$_V$1.2_IQM+FGF11A (E). Statistical comparisons are given in Table 3.

component (A$_s$) following a single depolarizing pulse was essentially identical between FGF14A and the 2Q mutant N terminus (Fig. 8 G), suggesting that the rate of entry into LTI was identical between FGF14A and FGF14A_2Q. Although, except for FGF14A_2Q, none of the other FGF14A mutants exhibited any hint of LTI, overall there was a trend for a somewhat faster onset of inactivation than that of WT Na$_V$1.2 alone (Fig. 8 H), consistent with effects of the various FGF-A homologues (Fig. 8 H). However, the slow component of recovery from inactivation FGF14A_2Q was about threefold faster than that of native FGF14A, suggesting a faster unbinding rate of the FGF14A_2Q N terminus from its position of inactivation (Fig. 8 I and Table 4). Overall, the lack of change in inferred rate of inactivation onset and the faster recovery corresponds approximately to a threefold weaker apparent binding affinity based on the presumed $k_{off}/k_{on}$ rates. This suggests that careful measurement of the time course of recovery from LTI can be of value in mapping molecular determinants for inactivation domain interactions that might point the way to better understanding of compounds of therapeutic value. In previous work, estimates of slow recovery from LTI were made on various FGF13A mutant N termini (Venkatesan et al., 2014), although associated estimates of effects on forward rates of development of LTI were not made. For the FGF13A-R11Q/R17Q mutant, visual inspection of the slow development of LTI suggests that, whereas the fractional amplitude of the slow component was about 0.5 for Na$_V$1.6+FGF13A, in the R11Q/R17Q mutant the fractional amplitude of the slow component was about 0.2

(Venkatesan et al., 2014). These earlier results together with the present observation on FGF14A_2Q point out that, with appropriate protocols, it may be possible to distinguish molecular determinants that may influence association from those affecting dissociation of FGF-A N termini from their sites of inactivation.

To complete this evaluation of the mutant constructs, we note that GV curves are relatively unaffected in these constructs, except for some unusual aspects of the LI/AA currents reflecting perhaps enhanced persistent current (Fig. 9, A–C; and Table 5). Inactivation time constants are slowed in all tested constructs except for FGF14A_ΔNT (Fig. 9, D–F; and Table 5). Finally, all constructs shift SSI to more positive voltages in most cases about 10–15 mV (Fig. 9, G–I), while the ΔNT construct produces the weakest shift (Table 5). The shifts produced by mutated N termini occur whether or not a construct actually produces LTI. This is consistent with the idea that FGFs, including both the A and the B isoforms, although perhaps not to the same extent, can allosterically influence the SSI equilibrium of the Na$_V$1.2 channel, independent of whether LTI can occur. Although the mean values for effects of FGF14B suggest that it also mimics the FGF14A gating shifts and changes in the slope of the SSI relationship, the shift with FGF14B appears less than that of FGF14A, generally consistent with previously published observations (Goldfarb, 2024). The ΔNT construct reflects the complete removal of all NT residues, while both A and B isoforms contain over 60 residues. Our results with ΔNT are in most cases indistinguishable from WT Na$_V$1.2 alone. As such, our results do

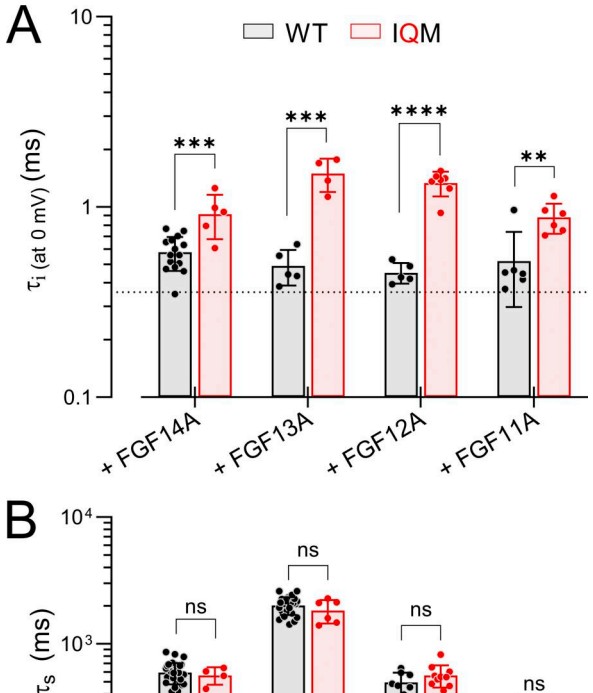

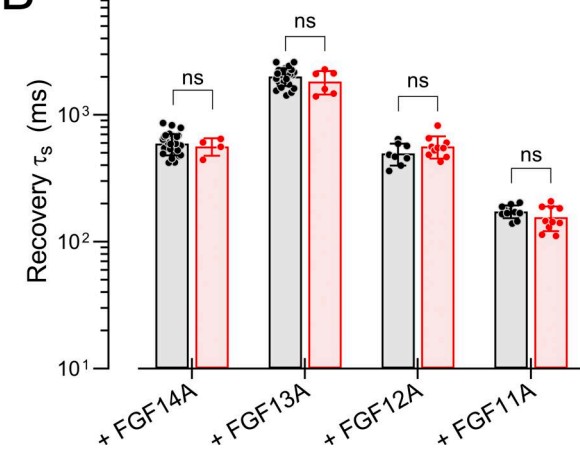

Figure 7. **Comparisons of effects of FGF-A isoforms on WT vs. IQM Na_V1.2 inactivation onset and recovery.** **(A)** Comparison of effect of FGF-A-isoforms on inactivation $\tau$ at 0 mV with mean ± SD for each current along with values from individual patches. Dotted line is control $\tau_i$ for Na_V1.2 alone. **(B)** Comparisons of slow recovery time constants (at −80 mV) for different FGFA isoforms in association with Na_V1.2 or Na_V1.2_IQM. Statistical comparisons (means ± SD, $N$, P values) are in Table S5. **P < 0.01, ***P < 0.001, and ****P < 0.0001.

not exclude the possibility that FGF14_ΔNT does not express and/or assemble with Na_V1.2.

These results with FGF14A mutations support the idea that there are features both in the FGF14A N terminus and also in the FGF14B N terminus that shift SSI curves to the right and also slow the onset of intrinsic fast inactivation. This occurs independent of the presence of any LTI. Furthermore, results with FGF14A_2Q highlight the value of careful delineation of the time constant of recovery from LTI and the slow component amplitude following a single depolarization step as tools for mechanistic probing of FGF-A interactions with Na_V channels.

**An allosteric effect of FGFs on intrinsic fast inactivation reconciles data with the LTI model**

Several observations presented above cannot be reconciled with the basic LTI model of FGF-A isoform action (Scheme 2). First, coexpression of Na_V1.2 with any FGF-A homologue results in a

slowing of inactivation onset, whereas Scheme 2 predicts that inactivation with both intact intrinsic fast inactivation and LTI should be faster than in the absence of an FGF-A. Second, for FGF14A and FGF13A, in particular, the fractions of channels that enter LTI during a single depolarization are about 0.6 and 0.5, respectively, indicative that $k_{lti}$ in each case should be similar to $k_i$, while direct measurements of $k_{lti}$ using the Na_V1.2_IQM construct yields a slower value for $k_{lti}$ than the WT $k_i$ estimate. Third, the fast component of recovery from inactivation for Na_V1.2+FGF-A currents is faster than that for recovery from intrinsic fast inactivation alone. Fourth, the evaluation of mutant FGF14A N termini revealed that, even in the absence of LTI, rightward shifts in SSI curves and slowing of fast inactivation onset were observed, suggesting the FGF's mediate allosteric effects on Na_V inactivation, independent of LTI. In regards to the latter and as summarized by Goldfarb (Goldfarb, 2024), rightward shifts of about 10–20 mV in SSI curves are a shared feature of most studies of FGFs (Lou et al., 2005; Goldfarb et al., 2007; Laezza et al., 2007; Dover et al., 2010; Seiffert et al., 2022; Marra et al., 2023). Such shifts, albeit exhibiting some variability in magnitude, are generally observed with either A or B N-terminal isoforms, again suggesting that the shifts are not dependent on the development of LTI per se. The physical basis for FGF-mediated shifts in SSI is not fully understood (Goldfarb, 2024).

Here, we evaluate the idea that the shifts in SSI and slowing of onset of intrinsic fast inactivation that occurs with FGF's may both arise from allosteric changes in transitions involved in the development of intrinsic fast inactivation. Specifically, we ask whether the quantitative measurements can be explained by a modified LTI model (Scheme 3) in which a key feature of the effects of FGF occurs independent of LTI per se and serves to slow intrinsic inactivation, thereby shifting SSI curves.

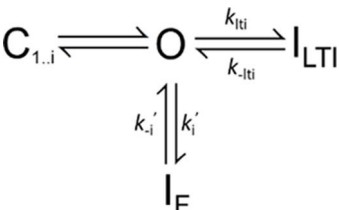

Scheme 3.

Scheme 3 differs from Scheme 2 by the idea that the presence of an FGF is altering the properties of the O⟷I_F equilibrium, potentially resulting in altered rates of fast inactivation onset and recovery, $k_i'$ and $k_{-i}'$. We assume the following:

(1) After a strong depolarization, the fraction of channels that will have entered LTI states is defined by $A_s = k_{lti}/(k_{lti} + k_i')$, where $k_i'$ is the allosterically modified rate of fast inactivation.

(2) Second, $k_{lti}$ and $k_{-lti}$ are unchanged by the presence or absence of the native fast inactivation process and are revealed by the time constant of inactivation onset and recovery observed with the Na_V1.2_IQM construct. This assumption seems appropriate, since the time constants of slow recovery ($k_{-lti}$) seem largely indistinguishable whether measured for WT or IQM.

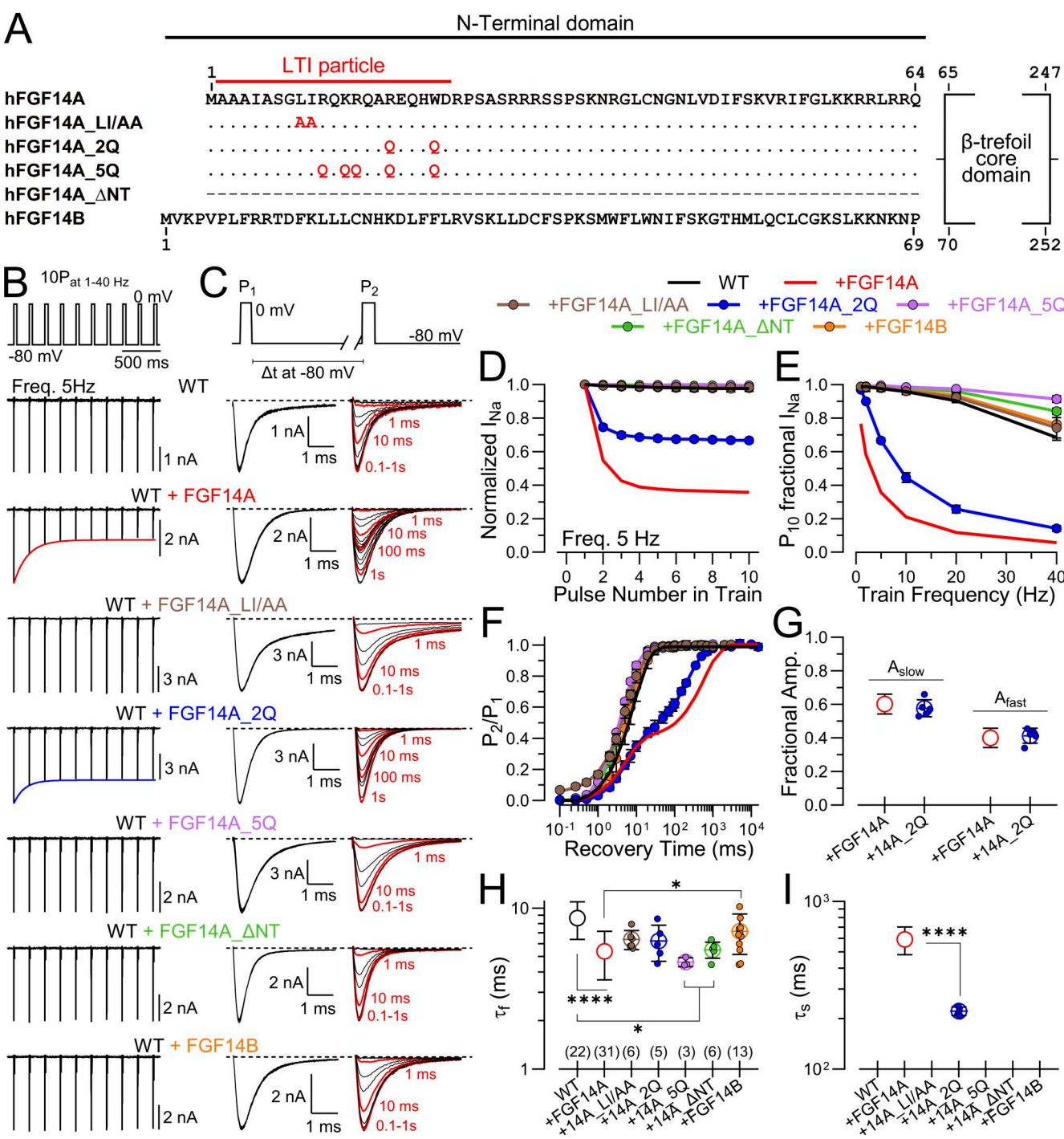

Figure 8. **Evaluation of LTI mediated by FGF14A N-terminal mutations. (A)** Schematic of FGF14A mutation sequences. **(B)** 10 pulse 5 Hz train-mediated inhibition for various FGF14A-mutated constructs. **(C)** Example currents from paired-pulse protocol for recovery from inactivation for various constructs. **(D)** $Na_V$ attenuation during 5 Hz 10P trains. **(E)** Fractional attenuation of $I_{Na}$ for 10th pulse in trains of different frequencies. **(F)** Time course of recovery from inactivation for the indicated constructs, along with best fits of single or double exponential functions. **(G)** Comparison of fraction of fast and slow amplitude components for FGF14A and FGF14A_2Q. **(H)** Comparison of fast recovery from inactivation for different constructs. **(I)** Comparison of slow recovery from inactivation for FGF14A and FGF14A_2Q. Statistical comparisons are given in Table 4. *P < 0.05 and ****P < 0.0001

From the assumption that $A_s$ directly reflects the differential rates of entry into $I_{LTI}$ and $I_F$, rearranging $A_s = k_{lti}/(k_i' + k_{lti})$ yields $k_i' = k_{lti}(1 - A_s)/A_s$. Table 6 lists various measured values in columns 2–6, while column 7 lists $k_i$ at the top for $Na_V1.2$ alone, and then the calculated modified $k_i'$ for each FGF-A homologue.

Column 8 lists the ratio of $k_i/k_i'$, indicating the fold-change in slowing of the intrinsic fast inactivation rate. This change in rate is the factor necessary to account for the fractional entry into LTI. Columns 9–10 list the experimentally measured values for $A_s$ and inactivation $\tau_i$ for each construct. Then based on the rate

Table 4. **Properties of onset and recovery from inactivation for FGF14A mutants and isoforms**

| Constructs | Recovery from inactivation at −80 mV | | | | | | |
|---|---|---|---|---|---|---|---|
| | $\tau_f$ (ms) | P value | $\tau_s$ (ms) | P value | $A_s$ | P value | N |
| $Na_V1.2$ | 8.7 ± 2.3 | | | | | | 22 (16) |
| +FGF14A | 5.4 ± 1.8 | <0.0001 [WT] | 592.9 ± 111.5 | | 0.60 ± 0.06 | | 31 (18) |
| +FGF14A_LI/AA | 6.4 ± 0.9 | 0.11 [WT] >0.99 [14A] | | | | | 6 (2) |
| +FGF14A_2Q | 6.3 ± 1.6 | 0.12 [WT] >0.99 [14A] | 221.8 ± 12.7 | <0.0001 [14A] | 0.58 ± 0.05 | 0.35 [14A] | 5 (2) |
| +FGF14A_5Q | 4.6 ± 0.3 | 0.008 [WT] >0.99 [14A] | | | | | 3 (2) |
| +FGF14A_ΔNT | 5.5 ± 0.6 | 0.005 [WT] >0.99 [14A] | | | | | 6 (3) |
| +FGF14B | 7.2 ± 2.0 | 0.27 [WT] 0.05 [14A] | | | | | 13 (5) |

Statistical analyses were performed using one-way ANOVA, followed by Bonferroni's multiple comparisons test. Statistical analyses of parameters from slow component between FGF14A and FGF14A_2Q were performed using unpaired $t$ test with Welch's correction.

of onset of LTI and the allosterically modified rate of intrinsic fast inactivation, we recalculate $A_s$ and the expected $\tau_i$ for the derived $k_i'$ (modified LTI prediction; columns 11–12). Since the $k_i'$ value was derived based on measured values of $A_s$, this regenerates the identical $A_s$ value. However, the model-calculated $\tau_i$ values (column 12: $\tau_i = 1,000/(k_{lti}+k_i')$) are being compared with experimentally measured $\tau_i$ values (column 10) obtained independently from other measurements. Although $\tau_i$ values predicted by the modified LTI model differ a bit from the experimentally measured inactivation time constants for the $Na_V1.2$+FGF-A homologues, they are within 1 SD of the measured values. When we calculate a predicted $A_s$ and $\tau_i$ based on a standard LTI model in which the intrinsic fast inactivation rate is not modified by the presence of an FGF (columns 13–14), we find in all cases that the estimated fraction of channels that enter LTI based on a single depolarization is markedly underestimated. Furthermore, the predicted inactivation time constant from the contributions of intrinsic fast inactivation and LTI are predicted to be much faster than inactivation arising from $Na_V1.2$ alone, which deviates substantially from experimental observations.

The proposal that intrinsic fast inactivation is slowed by FGFs also provides a potential explanation for the rightward shifts in the SSI curves (Table 6; column 3). It should also be noted that FGF14B also slows inactivation onset (column 2) in a fashion similar to FGF14A, consistent with the idea that these effects on inactivation onset are independent of the LTI process itself, although influencing it.

One aspect of the kinetic changes that occur with the FGFs that has not been evaluated here pertains to the time constant of recovery from intrinsic fast inactivation (Table 6; column 5). This presumably reflects $k_{−i}$ in the LTI Scheme 2. Whereas $Na_V1.2$ alone recovers at −80 mV with a time constant of 8.7 ms, when coexpressed with the different FGF-A homologues, the measured fast component of recovery is faster. We would suggest that this may be another manifestation of the allosteric effect produced by an FGF on intrinsic fast inactivation. Thus, in

accordance with the idea that entry and exit from fast inactivation can be crudely approximated by a two-step process ($O \leftarrow\rightarrow I_F$), the presence of an FGF slows the rate of entry into $I_F$ ($k_i > k_i'$) and increases the rate of recovery from $I_F$ ($k_{−i}' > k_{−i}$). Both effects destabilize occupancy in $I_F$ and might be expected to contribute to persistent $Na_V$ current as channels are less likely to dwell in $I_F$. We have not explored that possibility, but the simultaneous presence of $I_{LTI}$ occupancy would diminish the likelihood of detecting persistent current.

On balance, these considerations indicate that an allosteric LTI model better approximates both the fractional entry into LTI of all the FGF-A homologues and also the observed time constants of inactivation onset. In addition, it may explain the rightward shifts in SSI curves and the faster time constants of the fast component of recovery during the double exponential recovery from LTI.

### Differences among A-homologues in entry and recovery from LTI during pulse trains

We now turn to an evaluation of the use-dependent changes in $Na_V$ availability that occur as a consequence of LTI mediated by each FGF-A homologue. Irrespective of the specific LTI model, a consequence of competition between entry into $I_F$ and $I_{LTI}$ is that, during trains of action potentials or depolarizing steps, the fraction of $Na_V$ channels that occupy the LTI-inactivated state(s) will increase (Dover et al., 2010; Martinez-Espinosa et al., 2021a; Martinez-Espinosa et al., 2021b). This arises, since, during any recovery interval between sequential stimuli, a larger fraction of $I_F$-inactivated channels will recover from inactivation than the fraction of LTI-inactivated channels that recover. Thus, channels that recover from $I_F$ inactivation become available for entry into LTI during each subsequent stimulus. We reasoned that, as a consequence of the differential rates of entry and exit from LTI exhibited by different FGF-A homologues, accumulation in LTI during pulse trains might differ among FGF-A homologues. Therefore, for WT $Na_V1.2$ without and with each FGF-A

**Figure 9. Mutations in FGF14A N terminus that remove LTI still produce shifts in SSI and slow intrinsic fast inactivation. (A)** GV curves for the indicated FGF14A constructs. **(B and C)** $V_h$ and $z$ values from fits of GV curves to individual cells. **(D)** Inactivation time constants at different voltages for mutant FGF14A constructs and FGF14B. **(E)** Example inactivation time courses for the indicated constructs plotted on a log scale, highlighting that most constructs produce slower inactivation onset compared with WT FGF14A, except for FGF14A_ΔNT. **(F)** Comparison of inactivation time constants at 0 mV, highlighting that N-terminal manipulations do not alter FGF14-mediated slowing of inactivation onset, except for complete deletion of NT (ΔNT). **(G)** Fractional availability curves for constructs with different N-terminal manipulations. **(H and I)** $V_h$ and $z$ values for SSI for different N-terminal manipulations. Statistical comparisons with P values are summarized in Table 5. *P < 0.05, **P < 0.01, and ****P < 0.0001.

homologue, a single pulse and then 10-pulse trains were applied at different frequencies to probe the time course of recovery from inactivation (Fig. 10). Examples of development of LTI induced by a 10-pulse train applied at 5 Hz show that the extent of LTI by the 10th pulse varies appreciably among FGF-A homologues (Fig. 10 A). At the end of each train (righthand traces in Fig. 10 A), we also monitored recovery from inactivation to assess how the amplitude of fast and slow components of recovery and the time constants might vary. Following inactivation of WT Na$_V$1.2 channels produced by 10 pulses at 5–40 Hz trains, recovery from inactivation reflects exclusively intrinsic fast recovery (Fig. 10 B). However, for each FGF-A homologue, as train frequency is increased up to

40 Hz, there is a corresponding increase in the fraction of channels that recover via the slow recovery pathway (Fig. 10, C and D; and Table S6). Whereas the rank order of the fractional entry into slow recovery ($A_s$) for a single depolarizing pulse is FGF14A > FGF13A~FGF11A > FGF12A, after the 40 Hz train the rank order for occupancy in LTI becomes FGF14A > FGF13A > FGF12A > FGF11A (Fig. 10 D). Given the slow rate of recovery from LTI for FGF13A, one might expect that, for currents with FGF13A, the fraction of channels in LTI at the end of the 40 Hz train might be much larger than observed for other FGF-A homologues. In line with aspects of FGF13A mentioned above, we suggest the smaller than expected apparent occupancy of Na$_V$1.2+FGF13A channels in LTI may

Table 5. **Parameters from activation GVs, inactivation time constants, and SSI curves for FGF14A mutant constructs and FGF14B**

| Constructs | GV curve | | | | Inactivation $\tau$ (0 mV) | | |
| --- | --- | --- | --- | --- | --- | --- | --- |
| | $V_h$ (mV) | P value | z (e) | P value | $\tau_i$ (ms) | P value | N |
| $Na_V1.2$ | −17.1 ± 6.5 | | 4.35 ± 1.49 | | 0.36 ± 0.09 | | 14 (8) |
| +FGF14A | −20.3 ± 4.2 | 0.69 [WT] | 4.47 ± 0.88 | >0.99 [WT] | 0.58 ± 0.12 | <0.0001 [WT] | 15 (10) |
| +FGF14A_LI/AA | −25.3 ± 2.2 | 0.01 [WT] 0.07 [14A] | 4.81 ± 0.81 | 0.98 [WT] 0.99 [14A] | 0.57 ± 0.03 | <0.0001 [WT] >0.99 [14A] | 4 (3) |
| +FGF14A_2Q | −22.0 ± 1.6 | 0.15 [WT] 0.84 [14A] | 5.20 ± 0.68 | 0.64 [WT] 0.57 [14A] | 0.47 ± 0.03 | 0.009 [WT] 0.03 [14A] | 4 (2) |
| +FGF14A_5Q | −22.6 | | 5.65 | | 0.78 | | 1 (1) |
| +FGF14A_ΔNT | −19.5 ± 4.6 | 0.97 [WT] >0.99 [14A] | 4.72 ± 1.30 | 0.99 [WT] 0.99 [14A] | 0.30 ± 0.05 | 0.61 [WT] <0.0001 [14A] | 5 (3) |
| +FGF14B | −22.6 ± 3.2 | 0.09 [WT] 0.64 [14A] | 4.94 ± 0.55 | 0.79 [WT] 0.60 [14A] | 0.64 ± 0.09 | <0.0001 [WT] 0.72 [14A] | 12 (4) |

| | SSI curve | | | | | |
| --- | --- | --- | --- | --- | --- | --- |
| | $V_h$ (mV) | P value | z (e) | | P value | N |
| $Na_V1.2$ | −62.4 ± 3.5 | | 4.51 ± 0.50 | | | 22 (16) |
| +FGF14A | −48.2 ± 2.6 | <0.0001 [WT] | 5.85 ± 0.64 | | <0.0001 [WT] | 31 (18) |
| +FGF14A_LI/AA | −47.4 ± 3.0 | <0.0001 [WT] >0.99 [14A] | 5.54 ± 0.37 | | 0.007 [WT] >0.99 [14A] | 6 (2) |
| +FGF14A_2Q | −47.6 ± 1.2 | <0.0001 [WT] >0.99 [14A] | 6.36 ± 0.78 | | <0.0001 [WT] >0.99 [14A] | 5 (2) |
| +FGF14A_5Q | −46.2 ± 2.2 | <0.0001 [WT] >0.99 [14A] | 5.62 ± 1.07 | | 0.06 [WT] >0.99 [14A] | 3 (2) |
| +FGF14A_ΔNT | −57.5 ± 1.2 | 0.004 [WT] <0.0001 [14A] | 5.19 ± 0.40 | | 0.23 [WT] 0.21 [14A] | 6 (3) |
| +FGF14B | −52.9 ± 3.2 | <0.0001 [WT] <0.0001 [14A] | 5.23 ± 0.74 | | 0.02 [WT] 0.04 [14A] | 13 (5) |

Statistical analyses were performed using Welch's ANOVA test, followed by Dunnett's T3 multiple comparisons test (for GV data) or one-way ANOVA, followed by Bonferroni's multiple comparisons test (for SSI data).

reflect a slow reduction of the fraction of channels that enter into LTI that occurs with time after the onset of whole-cell recordings with FGF13A. For comparison, we undertook similar evaluation of use dependence for FGF-A homologue-mediated inactivation of $Na_V1.2$_IQM channels (Fig. S1). For such currents, a single depolarization was almost sufficient to reach a full steady-state level of reduction in $Na_V$ availability (Fig. S1, A and B), while the steady-state level of reduction at each train frequencies correlated well with the fractional recovery of channels during the interpulse interval expected based on the slow recovery time constants for each FGF-A homologue (Fig. S1 C).

Despite the substantial increase in the fraction of channels that reside in LTI with increases in train frequency for all FGF-A isoforms, over the range of tested train frequencies, the time constants of slow recovery following either a single pulse or a train exhibit little change (Fig. 10 E). This is consistent with the idea that the recovery time constants reflect the specific dissociation properties of a given FGF-A homologue from its binding site on $Na_V1.2$. In contrast, the rate of fast recovery following inactivation of an FGF-A–associated $Na_V1.2$ current is faster than that for WT $Na_V1.2$ alone (Table 2). However, as train frequency is increased, the fast time constant of recovery becomes slower, reaching values at the end of 40 Hz trains similar to the fast time constant of recovery for $Na_V1.2$ without an associated FGF (dotted lines on Fig. 10 E and Table S6). We have no definitive explanation for why the fast recovery time constant may change with train frequency. However, we would propose that the initial faster time constant of recovery following single pulses or lower frequency trains may relate to the allosteric interactions between the presence of FGFs and the intrinsic fast inactivation process affecting the O↔$I_F$ equilibrium as noted above, a phenomenon that itself may exhibit use-dependent reduction during trains.

The above results support the view that use-dependent increases in occupancy in slow recovery $I_{LTI}$ states arise because channels in LTI are unable to appreciably recover from inactivation between depolarizations, while fast recovery permits some fraction of the recovered channels to then be driven into LTI with each subsequent stimulus. This is the expectation for any model of LTI (Schemes 1, 2, and 3) embodied by two distinct, independent recovery pathways from $I_F$ and $I_{LTI}$.

**Table 6. Calculations used to determine allosteric effect of FGF-A homologues slowing intrinsic fast inactivation assuming modified LTI model**

| | Onset | | | Recovery | | | Inferred | | | Expt. | | Modified LTI prediction | | Standard LTI prediction | |
|---|---|---|---|---|---|---|---|---|---|---|---|---|---|---|---|
| | $\tau_i$ (ms) | SSI | $k_i$ (/s) | $k_{tti}$ (/s) | $\tau_r$ (ms) | $\tau_s$ (ms) | $k_i'$ (/s) | $\tau_i'$ (ms) | $k_i/k_i'$ | $A_s$ | $\tau_i$ (ms) | $A_s$ (LTI') | $\tau_i$ (ms) | $A_s$ (LTI) | $\tau_i$ (ms) |
| Na$_V$1.2 | 0.36 ± 0.09 | −62.4 | 2777.8 | | 8.7 ± 2.3 | | 2777.8 | 0.36 | | | 0.36 | | | | |
| +14A | 0.58 ± 0.12 | −48.2 | | | 5.4 ± 1.8 | 592.9 ± 111.5 | 724.6 | 1.38 | 3.83 | 0.60 | 0.58 | 0.60 | 0.55 | 0.28 | 0.26 |
| IQM+14A | 0.92 ± 0.24 | −44.4 | | 1087 | – | 565.5 ± 88.3 | | | | | | | | | |
| FGF14B | 0.64 ± 0.09 | −52.9 | | | 7.2 ± 2.0 | | 1562.5 | | 1.78 | | | | | | |
| +13A | 0.49 ± 0.10 | −49.0 | | | 6.3 ± 1.7 | 2017.6 ± 324.0 | 814.8 | 1.23 | 3.23 | 0.45 | 0.49 | 0.45 | 0.67 | 0.19 | 0.29 |
| IQM+13A | 1.50 ± 0.30 | −40.8 | | 666.7 | – | 1831.5 ± 383.5 | | | | | | | | | |
| +12A | 0.45 ± 0.06 | −51.8 | | | 5.0 ± 2.6 | 497.2 ± 97.7 | 1840.9 | 0.54 | 1.51 | 0.29 | 0.45 | 0.29 | 0.39 | 0.21 | 0.28 |
| IQM+12A | 1.33 ± 0.20 | −43.5 | | 751.9 | – | 565.4 ± 112.5 | | | | | | | | | |
| +11A | 0.52 ± 0.22 | −55.4 | | | 7.9 ± 2.7 | 173.1 ± 19.7 | 1446.3 | 0.69 | 1.92 | 0.44 | 0.52 | 0.44 | 0.39 | 0.29 | 0.26 |
| IQM+11A | 0.88 ± 0.16 | −43.8 | | 1136.4 | – | 155.8 ± 34.5 | | | | | | | | | |

To calculate the effective intrinsic fast inactivation rate, $k_i'$ when an FGF is associated with Na$_V$1.2, we assume that this modification is unrelated to the onset or recovery from LTI, but arises from an interaction perhaps of the trefoil domain or other part of the FGF N-terminal with the Na$_V$ subunit mediating an allosteric effect on inactivation. This arises independent of the LTI inactivation domain.

Column 1: construct designation

Column 2–7: measured values reported above

**Inferred rates and time constants for Allosteric LTI model.**

Column 8: Calculated $k_i'$ from $A_s = k_{tti}/(k_{tti} + k_i')$, rearranging yielding $k_i' = k_i' (1−A_s)/A_s$.

Column 9: $\tau_i = 1000/(k_i' + k_{tti})$, predicted inactivation tau in presence of FGF-A.

Column 10: fold-slowing of intrinsic fast inactivation rate.

**Expt. values**

Column 11: Measurements of $A_s$ following single depolarization.

Column 12: Measured time constant of inactivation for Na$_V$1.2+FGF-A homologue.

**Values predicted from allosteric LTI model, based on calculation of $k_i'$ based on As**

Column 13: $A_s$ from LTI* model.

Column 14: $\tau_i$ based on LTI* model.

**Values predicted from standard LTI model**

Column 15: $A_s$ from standard LTI model.

Column 16: $\tau_i$ based on standard LTI model.

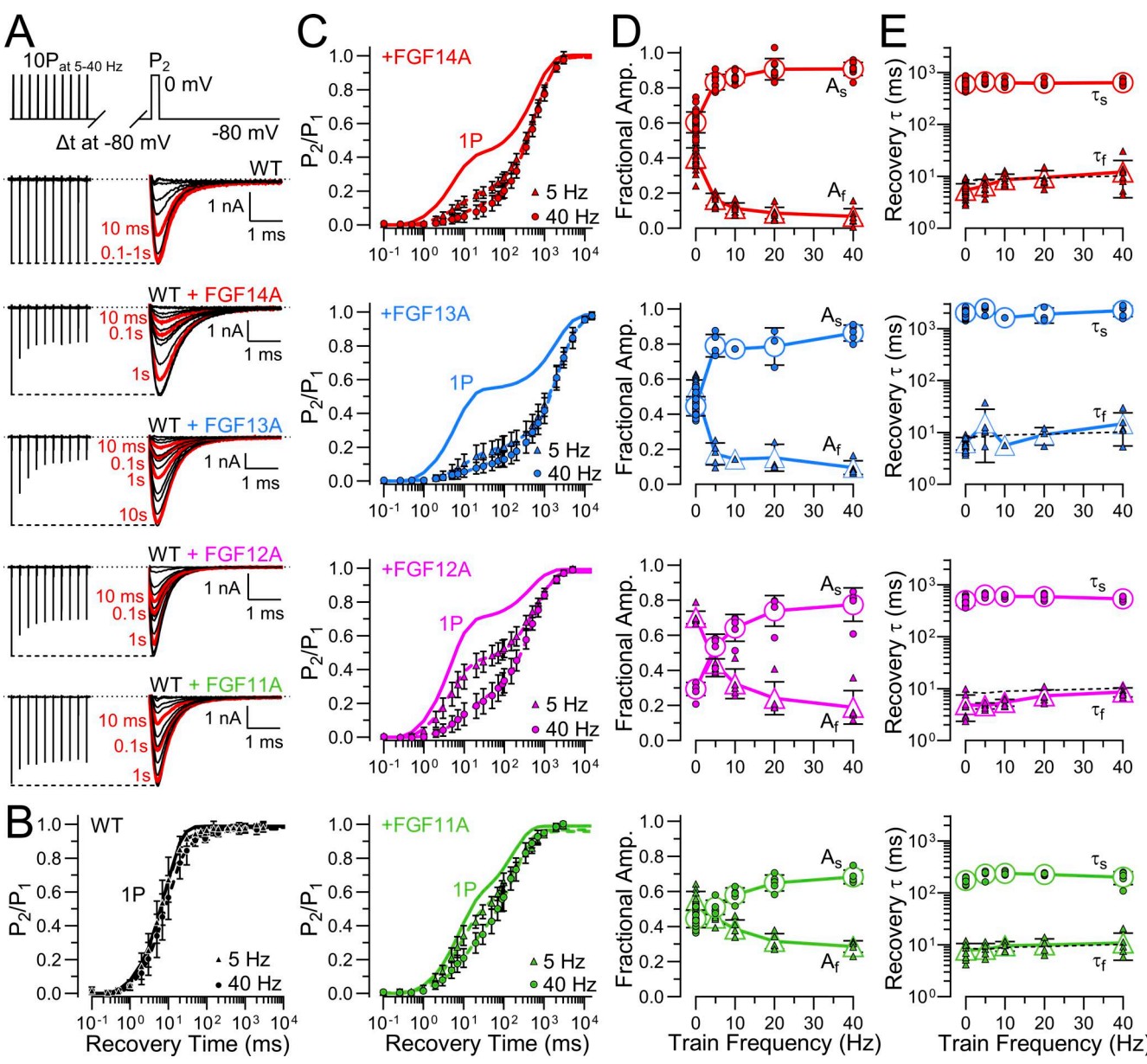

**Figure 10. Use-dependent accumulation in LTI produced by pulse trains varies among FGF-A isoforms. (A)** A 10-pulse train (10P) of 10-ms depolarizations to 0 mV applied at different frequencies (5–40 Hz, top panel) followed by recovery intervals at –80 mV from 0.1 ms to 5 s preceding another 25-ms step to 0 mV (P2) was used to examine recovery from inactivation mediated by FGF-A isoforms. Examples of $I_{Na}$ traces at 5 Hz pulse trains for Na$_V$1.2 (WT) alone and coexpressed with each FGF-A isoform are shown. Red traces show $I_{Na}$ current during P2 steps following recovery intervals of 1, 10, 100, and 1,000 ms. **(B and C)** Averaged time course of fractional recovery following a 10-pulse train at different frequencies for (B) WT alone and (C) in presence of each A-type FGF isoform, as indicated. Normalized $I_{PEAK}$ (P2 amp/P1 amp) from protocol, as in A, was used to assess recovery from inactivation following a single pulse (1P), or following 5 or 40 Hz 10-pulse trains. Lines show single exponential fits to data points (mean ± SD). Colored solid lines correspond to exponential fit of recovery from inactivation following a single step (1P) for Na$_V$1.2 alone (WT) (black) and WT coexpressed with FGF14A (red), FGF13A (blue), FGF12A (magenta), or FGF11A (green) in the corresponding panels. **(D)** Dependence of fractional amplitude of fast and slow recovery component on frequency of pulse trains for each construct along with values for individual cells. Fast and slow recovery components following a single step (1P) are plotted at "0 Hz." Data are expressed as mean ± SD (Table S6). **(E)** Dependence of fast and slow recovery time constants on frequency of pulse trains for each construct along with values for individual cells, with recovery time constants following a single step (1P) plotted at "0 Hz." Dotted line plots fast recovery for Na$_V$1.2 alone. Data are expressed as mean ± SD (Table S6).

## FGF-A-mediated use-dependent changes in Na$_V$1.2 availability during trains

Since decrements in $I_{Na}$ amplitude during pulse trains have been a common method for evaluating the presence and development of LTI, we next probed the FGF-A dependence of changes in Na$_V$

availability during trains, both with WT Na$_V$1.2+FGF-A's and Na$_V$1.2_IQM+FGF-A's. We hypothesized that the different kinetic features of LTI mediated by each FGF-A homologue might tailor a given Na$_V$+FGF-A channel complex to influence firing over specific frequency ranges. Plotting the average diminution

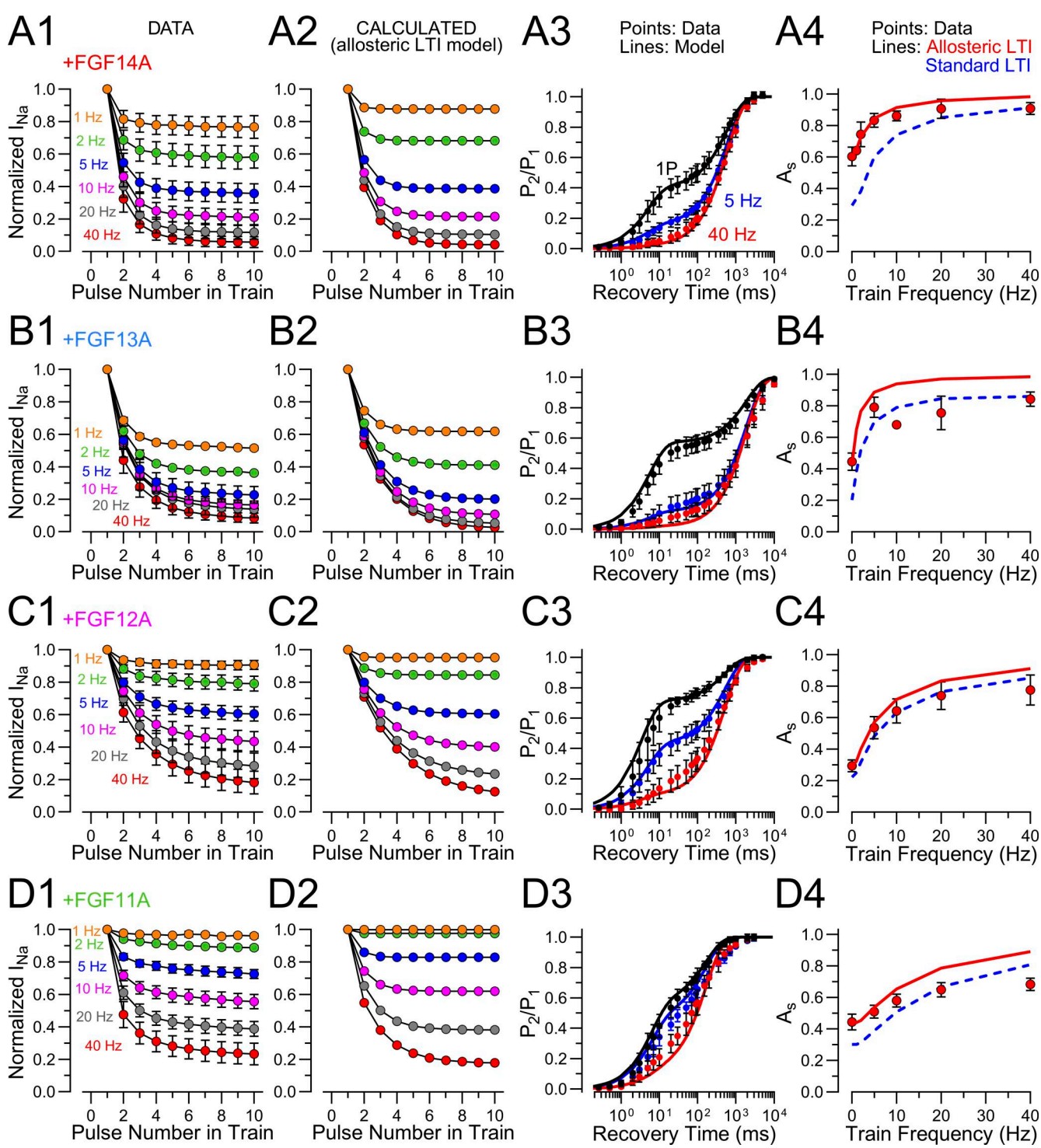

Figure 11. **Impact of FGF-A homologues on use-dependent changes in Na$_V$ availability. (A1)** Experimentally measured changes in peak I$_{Na}$ during a 10-pulse train applied at the indicated frequencies for Na$_V$1.2+FGF14A. **(A2)** Calculated diminution of Na$_V$1.2+FGF14A availability during 10-pulse trains applied at the indicated frequencies based on parameters for inactivation onset and recovery determined in accordance with modified LTI model (Scheme 3 and Table 6). **(A3)** Experimentally measured (points) and calculated recovery (lines) from inactivation for a single depolarization (1P) and also following a 10-pulse train at 5 Hz (blue) and a 10-pulse train at 40 Hz (red) for Na$_V$1.2+FGF14A. **(A4)** Experimentally measured (points) and calculated fractional A$_s$ for 1P recovery and then at the end of each of 10-pulse trains applied at the indicated frequencies with lines reflecting predictions from allosteric LTI model (red) and standard LTI model (dashed blue). **(B1–B4)** As for panels in A, but determined for parameters derived from Na$_V$1.2+FGF13A. **(C1–C4)** Calculated use dependence based on parameters derived from Na$_V$1.2+FGF12A. **(D1–D4)** Calculated use-dependence based on parameters derived from Na$_V$1.2+FGF11A.

in $I_{Na}$ amplitude during a 10 pulse train applied at frequencies from 1 to 40 Hz (Fig. 11, A1–D1) highlights the differential impact of each FGF-A homologue on $Na_V$ availability. For WT $Na_V1.2$ alone, diminution in peak $I_{Na}$ was minimal at 5 Hz (Fig. 10, A and B), while, with 5 Hz trains, peak amplitude with FGF-A homologues was reduced in order of FGF13A (~75%) ~FGF14A (~65%) > FGF12A (~40%) > FGF11A (~25%) (average reductions from Fig. 11). In addition to differences in the steady-state level of availability at different train frequencies among FGF-A homologues, differences in the number of pulses required to reach steady state also can also be seen.

We hypothesized that this use dependence should be explainable within the context of the kinetic transitions of the LTI model. Armed with measurements of the rates of entry ($k_{lti}$) and exit ($k_{-lti}$) from LTI (Table 6) and the inference of a potential allosteric effect of FGF's on $k_i$ allowing calculation of a modified $k_i'$ (Table 6), we have asked to what extent an allosteric model of LTI (Scheme 3) can approximate use-dependent decrements in $I_{Na}$. The basic strategy (see Materials and methods) was to calculate the fractional $Na_V$ availability for a series of 10 depolarizations (P1 to P10), beginning with 100% availability prior to P1, with depolarizations separated by a given recovery interval (1,000, 500, 200, 100, 50, and 25 ms) during which some fraction of channels in $I_{LTI}$ or $I_F$ differentially recover to availability. It is assumed that, during each depolarization, all available channels become inactivated in accordance with the differential rates of $k_i'$ and $k_{lti}$, i.e., $A_s = k_{lti}/(k_i' + k_{lti})$ with fractional recoveries from $I_{LTI}$ or $I_F$ defined by the respective recovery rates, $k_{-lti}$, and $k_{-i}$. From this procedure, fractional use-dependent reductions were calculated for $Na_V1.2$ in association with each of the FGF-A homologues at different train frequencies (Fig. 11, A2–D2). Although these calculated use-dependent changes in $Na_V$ availability do not precisely recapitulate the experimental data, they do a reasonable job of capturing the use-dependent reduction mediated by each of the FGF-A homologues. Based on the kinetic constants for $k_{-i}$ and $k_{-lti}$ and calculated changes in $A_s$ and $A_f$ during pulse trains, we compared the calculated time course of recovery from inactivation with the experimental time courses of recovery from inactivation (Fig. 11, A3–D3) for recovery after a single pulse, after a 5 Hz pulse train, and after a 40 Hz pulse train. Again the general correspondence supports the view that the use-dependent changes in occupancy fit well within the contact of the modified LTI model. Finally, we calculated the fractional occupancies of channels in $I_F$ ($A_f$) and $I_{LTI}$ ($A_s$) following the 10th pulse at each train frequency for each FGF-A homologue (Fig. 11, A4–D4) and compared this with $A_s$ values calculated either with the modified LTI model ($k_i$ modified by presence of an FGF-A) and with the standard LTI model ($k_i$ not modified by presence of FGF-A). For the FGF14A data set, which involved a larger number cells and included recovery information following 1 and 2 Hz trains, the modified LTI model provides a much better description of the use-dependent changes in occupancy in $A_s$ (Fig. 11 A4). In contrast, for FGF13A, for which recovery following trains at 1 and 2 Hz were not examined, much of the higher frequencies were better described by the standard LTI model (Fig. 11 B4). However, the standard model fails to account for the observation that a fraction of about 0.45 of the

$Na_V1.2$+FGF13A channels enter LTI following a single depolarizing step. As mentioned previously (also see Materials and methods), we suspect the low estimates of $A_s$ at high train frequencies for FGF13A reflects the rundown of LTI during long recording times. For FGF12A, both models are reasonably congruent with the data, including similar estimates for the $A_s$ value after a single depolarization (Fig. 11 C4). This similarity likely arises largely in part since the presumed allosteric effect of FGF12A on $k_i$ (1.51; column 8, Table 6) is smaller than that for FGF14A (3.83), FGF13A (3.23), and FGF11A (1.92). Finally, for FGF11A, neither LTI model does a clearly superior job describing the frequency-dependence occupancy in $A_s$ following trains of different frequency (Fig. 11 D4). Given the relatively rapid recovery from FGF11A-mediated LTI and smaller separation from the fast recovery time constant, we would suggest that the kinetic constants describing FGF11A action in our experiments are not as well-defined as for the other A-homologues. For comparison, the frequency dependence of reductions in $I_{Na}$ during trains for $Na_V1.2\_IQM$+FGF-A's were determined (Fig. S2, A1–D1), with each current almost approaching a steady-state level of available after a single depolarization. The predictions of the LTI model with the $O\leftrightarrow I_F$ transition deleted (Fig. S2, A2–D2) generally correlated with experimental observations, although predicting a more immediate decrease to the steady-state level after the P1 step than observed in the data.

## Might different FGF-A homologues play a role in tuning cells to particular firing frequencies?

The previous results demonstrate differences among FGF-A homologues in terms of the extent to which different train frequencies reduce steady-state $Na_V$ availability. Fig. 12 A highlights the differences among FGF-A homologues in the extent of diminution in peak $I_{Na}$ amplitude during a 5 Hz 10 pulse, which matches well with the diminution predicted by the allosteric LTI model for each FGF-A homologue (Fig. 12 B). To evaluate the idea that each FGF-A may be suited to influence $Na_V$ current availability most effectively in a given frequency range, we replotted the fractional reduction in the P10-evoked $I_{Na}$ current as function of train frequencies (Fig. 12 C). To provide an empirical measure of the differences among FGF-A homologues, we fit the amplitudes of the P10 values as a function of frequency to $I_{P10}(f) = \exp(-f/f_{50})$, with $f$ representing different train frequencies, and $f_{50}$ yielding the frequency at which the $I_{P10}$ amplitude was reduced to half of the maximal value. Although the actual data values suggest that a baseline component would improve the fit, the equation used has the merit of providing a more comparable measure of $f_{50}$ value (as indicated by the horizontal dotted line). We also undertook a similar evaluation of the calculated $I_{P10}$ values based on the allosteric LTI model (Fig. 12 D), which we fit in a similar fashion. $f_{50}$ estimates were also generated for each individual cell (Fig. 12 E) with statistical comparisons between FGF homologues given in the figure legend. We then plotted the $f_{50}$ values measured experimentally for each FGF-A homologue and the corresponding $f_{50}$ values from the modified LTI model (Fig. 12 D) as a function of the measured slow recovery time constants (Fig. 12 F). When the estimates of $f_{50}$ for each FGF-A homologue are plotted as a

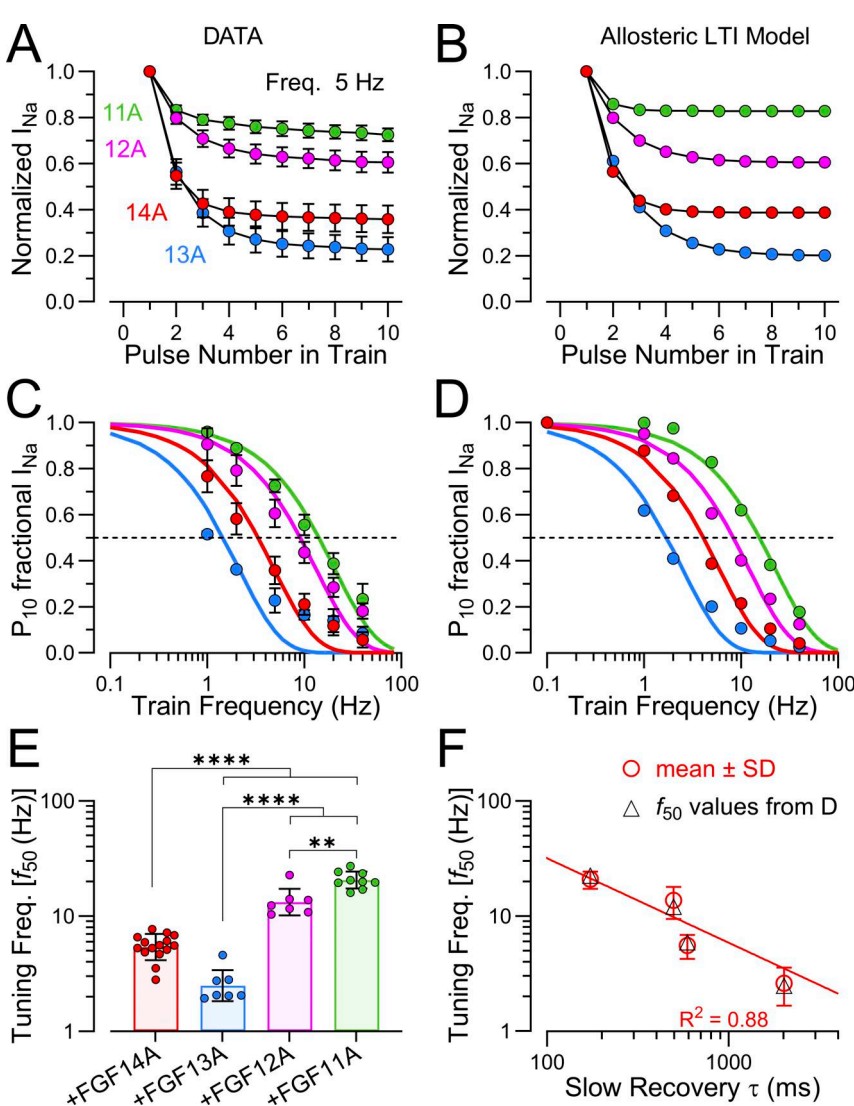

Figure 12. **Different FGF-A homologues differentially tune firing frequency. (A)** Replot of 5 Hz train data (Fig. 11, A1–D1) comparing dependence of peak $Na_V$ current amplitude on pulse number for each $Na_V1.2$+FGF-A combination. **(B)** Calculation of predicted reduction during 10-pulse trains applied at 5 Hz for each $Na_V1.2$+FGF-A combination based on rate constants for allosteric LTI model shown in Table 6. **(C)** Experimentally measured fractional amplitudes of P10-evoked current as a function of train frequency. Lines correspond to best fits of the $I_{P10}(f) = \exp(-f/f_{50})$, where $f$ is the train frequency and $f_{50}$ is the frequency at which fractional peak $Na_V$ is reduced to 0.5. Dotted line indicates level of half reduction of peak current. **(D)** Calculated P10-evoked current amplitudes based on the allosteric LTI model with fit as in D. **(E)** Plot of mean, SD, and individual values for fitted $f_{50}$ for different $Na_V1.2$+FGF-A pairings. Comparison of distributions used an ANOVA test followed by Tukey's multiple comparisons correction. For 14A vs. 13A, ****$P < 0.0001$; for 14A vs. 12A, ****$P < 0.0001$; for 14A vs. 11A, ****$P < 0.0001$; for 13A vs. 12A, ****$P < 0.0001$; for 13A vs. 11A, ****$P < 0.0001$; for 12A vs. 11A, $P = 0.008$. **(F)** $f_{50}$ values are plotted versus slow recovery time constant for different FGF-As. Red: mean $f_{50}$ values obtained from fits to individual cells used in panel C. Open symbols: $f_{50}$ values from fits of the calculated LTI model values (panel D). The fitted line has no physical meaning but highlights the trend of the $f_{50}$ vs. $\tau_s$ relationship. **$P < 0.01$.

function of the time constant for recovery from LTI (Fig. 12 F), there is a marked association of slower recovery from inactivation with lower frequency tuning. It is not unexpected that LTI would be expected to have a profound impact on $Na_V$ channel availability in cells in which FGF-A homologues are found and that this would be influenced by the rates of recovery from LTI. Although in native cells a variety of other factors will impact on firing rates, we suggest that the $f_{50}$ parameter has merit by defining a cell firing rate, where half the $Na_V$ channels (in this case, $Na_V1.2$) will be available for activation. For $f > f_{50}$, as availability falls below 50%, diminished firing may be the outcome.

## Discussion

Our results establish several new points regarding regulation of $Na_V$ channels by FGF-A homologues. First, despite extensive homology in the N-terminal sequences of FGF-A homologues, there are marked differences in both rates of onset and recovery from LTI among different FGF-A homologues. Second, the differences among FGF-A homologues in fractional entry into LTI

most likely arise in part from allosteric effects of each FGF-A homologue on the intrinsic rate of fast inactivation. Third, the overall functional effects of FGF-A isoforms likely arise from multiple distinct physical elements of the subunits. Fourth, as a consequence of the differences among FGF-A homologues both in rates of onset and rates of recovery from LTI, use-dependent changes in $Na_V$ availability vary among the four FGF-A homologues, potentially permitting each type of $Na_V1.2$+FGF-A complex to participate in tuning cells to particular firing frequencies. Here, we now discuss each of these points.

### Differences among FGF-A homologues in rates of recovery and onset of LTI-mediated inactivation

The results reveal clear differences in rates of recovery from LTI among FGF-A homologues. For each FGF-A homologue, the time constants of recovery either measured with native fast inactivation intact ($Na_V1.2$+FGF-A) or with fast inactivation removed ($Na_V1.2\_IQM$+FGF-A) were essentially indistinguishable. In accordance with the original conception of the origins of LTI (Goldfarb, 2012), the recovery time constants are proposed to reflect the dissociation rate of the unique FGF-A N-terminal

inactivation segment from a binding site on the associated Na_V channel that occludes ion permeation. At present, the occlusion proposal remains the best viable explanation for the underlying mechanism of LTI. Overall, the dissociation rates measured here vary over 10-fold among different FGF-A homologues, with FGF13A being the slowest at around 0.5/s and FGF11A being the most rapid at about 6/s. This suggests that the specific molecular interactions between N termini and binding sites differ markedly.

Concerning the onset of LTI, the quantitative estimates we have made depend to some extent on assumptions of the LTI model. However, irrespective of the underlying model, the measurements of onset of inactivation mediated by each FGF-A homologue on the Na_V1.2_IQM channel can be considered a direct measure of relative differences in intrinsic rates of binding of each N terminus to its position of occlusion. These values vary from 1,136/s for FGF11A to 666.7/s for FGF13A (Table 3) corresponding to a 1.7-fold range of forward rates, in contrast to the over 10-fold range of apparent dissociation rates.

Thus, the results indicate that each FGF-A homologue produces inactivation of Na_V1.2 channels that differs both in terms of rate of onset and rate of recovery. However, the rates of recovery exhibit the most marked differences among homologues and therefore might be expected to be the most significant contributor to differential effects of FGF-A homologues on cell excitability.

**LTI mediated by FGF-A homologues is best accounted for in part by allosteric effects on the rates of entry and exit from intrinsic fast inactivation**
Several aspects of the FGF-A effects are not readily accounted for by the simple models of LTI (e.g., Scheme 2). The most notable features not readily consistent with simple LTI are: (1) slowing of the macroscopic rate of inactivation onset when Na_V1.2 is coexpressed with any of the FGF-A homologues and (2) fractions of channel entry into LTI initiated by a single depolarization that are also not consistent with the slowing of macroscopic inactivation rates. In addition, other FGF effects that are not readily explained within the context of LTI, but may reflect important aspects of FGF action influenced by LTI, include rightward shifts in SSI curves and also increases in rates of fast recovery from inactivation.

We were able to account for all the above features of FGF-A action within the context of the LTI model by postulating that FGF's also modify the transitions governing the overall $O \leftrightarrow I_F$ equilibrium, thereby slowing inactivation onset ($k_i$) and speeding up recovery ($k_{-i}$). This led to calculation of $k_i'$, the allosterically modified rate constant of fast inactivation, when Na_V1.2 is associated with an FGF subunit. Both the changes in apparent time constant of inactivation in the presence of an FGF-A homologue and also the fractional entry into LTI during a single depolarization could be accounted for within the context of the modified LTI model, assuming each FGF-A homologue uniquely impacts on $k_i'$. Although the slowing of fast inactivation by the FGF-A homologues also potentially provides an explanation for the rightward shifts in SSI, one might then expect a correlation between the modification of $k_i'$ by

each FGF-A homologue and the shifts in SSI. However, no such correlation is observed.

Irrespective of whether an allosteric version of the LTI model is an adequate mechanistic description of the effects of FGF-A–mediated inactivation, whatever the underlying mechanism, it needs to account for all of the above features of the FGF-A effects. Although it might be argued that the allosteric version of the LTI model adds complexity that perhaps requires consideration of some alternative mechanistic proposal, that inclusion of an FGF effect solely on the $O \leftrightarrow I_F$ equilibrium is able to account for multiple distinct features of our results gives some confidence that the modified allosteric LTI model has merit. One potential value of the modified LTI model is that it may provide a more robust framework for quantitative evaluation of manipulations designed to better understand the LTI effects mediated by FGF-A homologues, specifically in terms of defining sequence elements that influence recovery from inactivation vs. onset of inactivation.

**Might slow inactivation impact on any of our results and interpretations**
In the Results, we acknowledged that the 500 ms prepulse used in our SSI protocols may have resulted in some slow inactivation but that it was unlikely to impact on our observations regarding shifts in SSI curves mediated by the FGF-A homologues. Furthermore, Fig. 10 B shows that even a 10 pulse 40 Hz train of steps to 0 mV produces minimal entry into slow inactivation in WT Na_V1.2 channels. Thus, LTI occurs under conditions for which slow inactivation does not occur. Given the rapidity of the LTI inactivation onset, that it results in essentially full inactivation of Na_V1.2 when intrinsic fast inactivation removed, and that it involves specific residues in the FGF-A N-terminal segment, the idea that LTI might arise from alteration of the intrinsic slow inactivation process seems unlikely. Although future experiments would be required to fully address this topic, we propose that, as is the case for intrinsic fast and slow inactivation, the occurrence of slow inactivation will occur in parallel with LTI.

**Functional effects of FGF-A homologues arise from multiple distinct physical elements of the subunits**
The results suggest that different aspects of the FGF-A subunit effects may arise from distinct physical elements of the subunits. First, as developed by the work from the Goldfarb lab (Dover et al., 2010; Goldfarb, 2012), some number of the penultimate residues of each FGF-A N terminus seem essential for the occurrence of LTI. However, an implication of the results with the FGF14-2Q mutant construct is that residues that may disrupt the forward rate of LTI may to some extent be distinguishable from those that can affect stability of the N terminus in its position of inactivation. Second, FGF14A mutant constructs, FGF14B, and different FGF-A homologues all produce a slowing of the onset of inactivation, irrespective of whether LTI occurs, and also produce an increased rate of recovery from fast inactivation, features of the allosteric effects on the $O \leftrightarrow I_F$ equilibrium. That the latter effects occur in the absence of LTI, in all FGF-A homologues, and also with FGF-B homologues suggest that sequences

defined within the N-terminal exon 1 probably do not impact on these aspects of FGF effects.

One approach to probing FGF–Na$_V$ interactions has been to build homology models to an existing structure of FGF13 in complex with the C-terminal domain of Na$_V$1.5 (Wang et al., 2012) to identify residues that may participate in defining potential protein–protein interactions between particular FGFs and partnered Na$_V$ subunits. In FGF13VY, residue R120 has been implicated in stability of association with the Na$_V$1.5 CTD (Gade et al., 2025). For FGF14B in association with Na$_V$1.6, a large set of residues have been probed as potential loci that may be involved in stability of interaction of FGF14 and Na$_V$1.6, including residues positioned throughout FGF14 downstream of the exon 1 residues. This includes K74 and I76, which occur near the beginning of the second exon (Ali et al., 2016), Y158 and V160 (Ali et al., 2016), and then more distal sequences (e.g., FLPK) closer to the C terminus (Singh et al., 2020; Singh et al., 2021; Arman et al., 2025). Both the K74/I76 pair of residues and the FLPK sequences are shared among FGF homologues. In a similar fashion, F189S in FGF14, a naturally occurring mutation identified in a Dutch family, may interfere with FGF14 association with hippocampal Na$_V$ channels and shift V$_h$ of SSI leftward about 5 mV (Laezza et al., 2007), again consistent with the idea that assembly is disrupted. Overall, the available results support the idea that various FGF residues, which may contribute to FGF:Na$_V$ interfaces, may hinder coassembly. Although results are not entirely consistent on this point, in many cases the V$_h$ of SSI does shift negatively in the absence of assembly with the FGF.

Although these considerations suggest that important determinants of interactions between FGF and Na$_V$ can be defined, this approach does not answer the question of how association with FGF is mediating the shifts in SSI. In principle, it might be possible to probe determinants of the SSI shifts by taking advantage of the robust properties of the slow time course of recovery from LTI to look for FGF elements that sustain normal LTI behavior while no longer permitting SSI shifts. Of course, a difficulty may be that the ability of FGF's to produce SSI changes may reflect the concerted effects of multiple distributed interactions that may preclude this type of targeted approach.

### Use-dependent changes in Na$_V$ availability mediated by FGF-A homologues may impact on tuning of cells for a particular useful range of firing frequencies

Use-dependent changes in Na$_V$ availability vary among the four FGF-A homologues as a consequence of differences both in rates of onset and also rates of recovery from LTI. Furthermore, the differential dependence on train frequency among FGF-A homologues in producing use-dependent decrements in Na$_V$ availability strongly supports the idea that each type of Na$_V$1.2+FGF-A complex is specifically suited to tune cells to particular firing frequencies. Thus, we propose that, for cells predominantly with Na$_V$+FGF13A channels, firing rates much in excess of 1–2 Hz would probably not be sustainable as Na$_V$ availability falls below 0.5. Our tests utilized a recovery potential of –80 mV, so in many cells with more depolarized resting potentials, defined by whatever constellation of conductances are present in the cell, even 1–2 Hz firing may become untenable. FGF14A would also be expected to

be associated with cells of relatively low frequency firing perhaps around 5 Hz. Relevant to this, in rodent adrenal CCs (Martinez-Espinosa et al., 2021a; Martinez-Espinosa et al., 2021b), FGF14A appears to underlie LTI of Na$_V$ current, arising primarily from Na$_V$1.3 in mouse CCs, but perhaps more Na$_V$1.7 in rat cells. CCs spontaneously fire at around 1–2 Hz, but with depolarizing stimuli AP firing can occur with an instantaneous frequency of about 10 Hz that quickly accommodates to loss of a sodium-dependent AP component (Solaro et al., 1995). Slow waveforms producing bursts can result in a series of action potentials that rapidly attenuate, consistent with the idea the repetitive activity at higher frequencies cannot be sustained (Martinez-Espinosa et al., 2014). FGF12A seems suitable for a role in cells that function in the 5–15 Hz range, while FGF11A might contribute to firing in cells "designed" to function even at 20–30 Hz range. As yet, our understanding of where FGF-A homologues are expressed in native cells remains limited, but only a few cell types have utilized tests that would clearly reveal the presence of an LTI inactivation mechanism. One such loci for which LTI clearly occurs is in slow pacemaker neurons of the raphe nuclei (Milescu et al., 2010; Navarro et al., 2020). Of course, cells could circumvent the firing rate limitations that FGF-A homologues may place on a given Na$_V$ current component by also expressing populations of Na$_V$ channels that lack associated FGF-A homologues.

### FGF-A isoforms in native cells

The LTI behavior conferred on Na$_V$ channels by FGF-A homologues is robust and has been clearly shown for Na$_V$1.6+FGF13A (Rush et al., 2006; Dover et al., 2010; Venkatesan et al., 2014) and Na$_V$1.6+FGF14A (Laezza et al., 2009). How widespread FGF-A–mediated LTI may be in native cells remains less clear. A challenge is that in many cells, both A- and B-isoforms may be present, even potentially involving multiple FGF homologues. In many native cells, specific protocols to test for the presence of LTI have not been reported or not been done. One standard test for the presence of an FGF-A–mediated inactivation process is the appreciable decay of peak Na$_V$ current amplitude with trains of pulses applied at modest frequencies (Rush et al., 2006; Laezza et al., 2009; Dover et al., 2010; Venkatesan et al., 2014). Perhaps better is application of a paired pulse protocol to specifically measure fast and slow components of recovery from inactivation. As described here, the slow time constant of recovery from LTI even at –80 mV and room temperature spans 150 ms (FGF11A) to 1,800 ms (FGF13A), a range often not explored in studies of native cells. The merit of the paired-pulse protocol is that it presumably directly reveals a specific molecular rate constant, that of dissociation of the inactivation segment from its binding site, whereas pulse trains only reveal the presence of LTI. In the few cases it has been examined, a 10 Hz train of 10 pulses can quickly identify the likely presence of LTI. In CA1 pyramidal neurons, a 10 pulse 20 Hz train results in an ~20% reduction of peak I$_{Na}$ (Venkatesan et al., 2014). In both rat and mouse CCs, a 10 pulse 10 Hz train results in an ~60% reduction in peak I$_{Na}$ (Martinez-Espinosa et al., 2021a; Martinez-Espinosa et al., 2021b). In raphe neurons, a slowly developing use-dependent attenuation of peak I$_{Na}$ to about 50% occurs with a 10 pulse ~10 Hz train (Milescu et al., 2010), with similar

effects ascribed to LTI subsequently described in dorsal raphe neurons (Navarro et al., 2020). Unfortunately, despite the distinctive features of LTI mediated by A-isoforms, most attention pertinent to FGFs seems to have focused on B-isoforms or simply on changes that arise from FGF KO without focus on what the predominant isoform might be.

## Data availability

All constructs generated in this work are available by request, and all data will be made available following any reasonable request.

## Acknowledgments

Olaf S. Andersen served as editor.

We thank Al George for providing the HEK cell line stably transfected with $Na_V\beta1$ and $Na_V\beta2$.

This work was supported in part by the National Institutes of Health (NIH) GM-118114 to C.J. Lingle. Y. Lorenzo-Ceballos received salary support from NIH-T32GM108539.

Author contributions: Y. Lorenzo-Ceballos: formal analysis, investigation, and visualization. P.L. Martinez-Espinosa: conceptualization, data curation, and investigation. X.M. Xia: data curation, investigation, methodology, validation, and writing—review and editing. C.J. Lingle: conceptualization, formal analysis, funding acquisition, methodology, project administration, resources, supervision, visualization, and writing—original draft, review, and editing.

Disclosures: The authors declare no competing interests exist.

Submitted: 18 February 2026

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

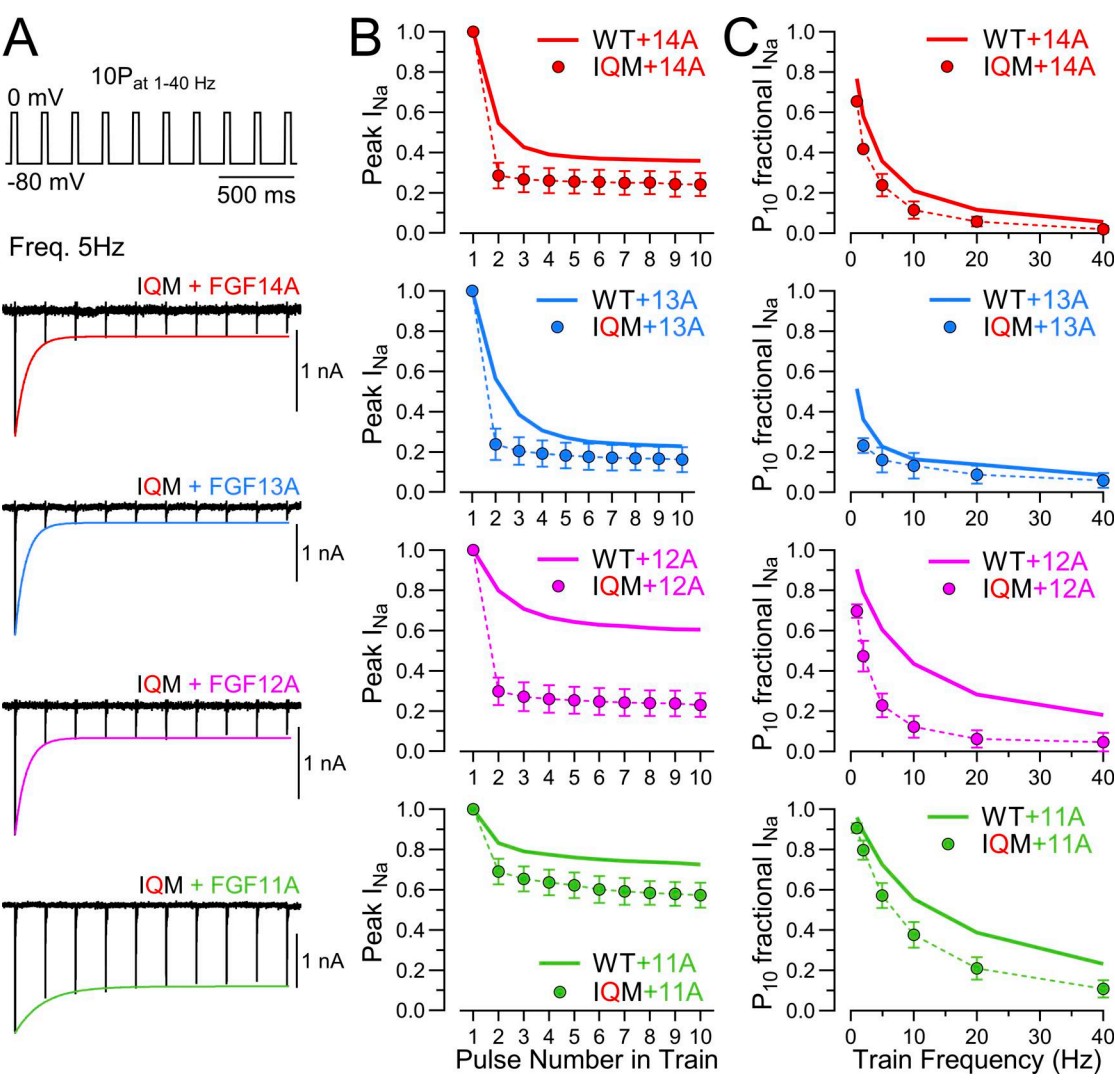

Figure S1. **Use-dependent changes in $I_{Na}$ availability mediated entirely by LTI during 10-pulse trains at different frequencies. (A)** 10-pulse trains applied at 5 Hz for WT $Na_V1.2$, and $Na_V1.2\_IQM$ with each of the four FGF-A homologues. **(B)** Averaged changes in peak $I_{Na}$ during 5 Hz trains for FGF-A homologues expressed with $Na_V1.2\_IQM$ (point) in comparison with channels with WT $Na_V1.2$ (lines). **(C)** Frequency dependence of reductions in $I_{P10}$ for FGF-A homologue expression with $Na_V1.2\_IQM$ in comparison with that for WT $Na_V1.2$ (lines).

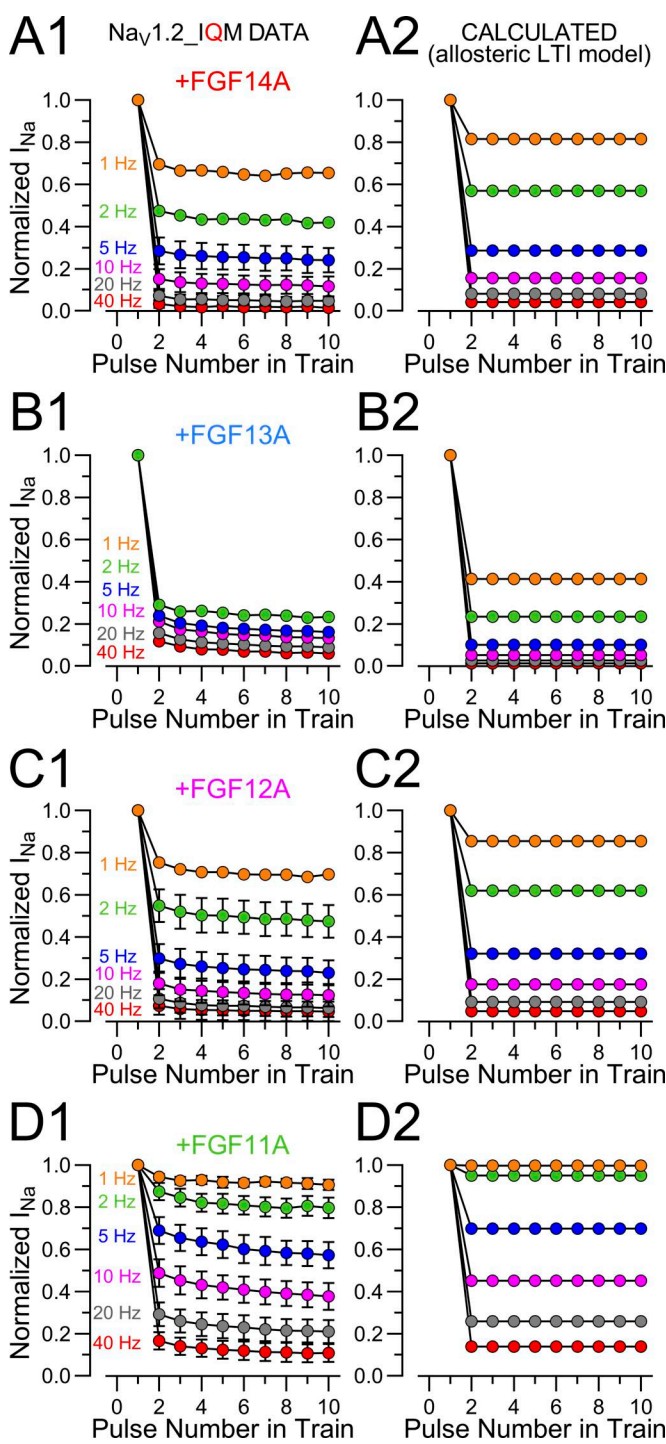

Figure S2. **The allosteric LTI model with fast inactivation removed is consistent with experimentally observed use-dependent changes in $Na_V1.2$ availability during trains. (A1)** A 10-pulse train of 10-ms depolarizations to 0 mV applied at different frequencies (1–40 Hz) was used to examine the cumulative inactivation of $Na_V1.2\_IQM$+FGF14A current as shown in Fig. S1. **(A2)** Predicted diminution during a 10-pulse train of depolarizations was calculated based on the allosteric LTI model for FGF14A for channels with fast inactivation removed. **(B1 and B2)** Measured and calculated diminution during pulse trains for $Na_V1.2\_IQM$+FGF13A currents. **(C1 and C2)** Measured and calculated diminution for $Na_V1.2\_IQM$+FGF12A currents. **(D1 and D2)** Measured and calculated diminution for $Na_V1.2\_IQM$+FGF11A currents.

Provided online are Table S1, Table S2, Table S3, Table S4, Table S5, and Table S6. Table S1 shows parameters determined from Boltzmann fits to GV curves for $Na_V1.2$ alone or when coexpressed with A-type FGF isoforms. Table S2 shows parameters

determined from Boltzmann fits to SSI curves for $Na_V1.2$ alone or coexpressed with A-type FGF isoforms. Table S3 shows activation parameters for $Na_V1.2\_IQM$ expressed alone or with FGF-A isoforms. Table S4 shows Boltzmann fit parameters to SSI curves for $Na_V1.2\_IQM$ alone or when coexpressed with A-type FGF isoforms. Table S5 shows effects of FGF-A homologues on $Na_V1.2$ and $Na_V1.2\_IQM$ GV curves, SSI, and taus. Table S6 shows use-dependent changes in FGF-A–mediated recovery from LTI.

**Table S1. Parameters determined from Boltzmann fits to GV curves for $Na_V1.2$ alone or when co-expressed with A-type FGF isoforms.**

| Constructs | GV curve | | | | Inactivation $\tau$ (0 mV) | | N |
|---|---|---|---|---|---|---|---|
| | $V_h$ (mV) | P value | z (e) | P value | $\tau_i$ (ms) | P value | |
| $Na_V1.2$ | -17.1 ± 6.5 | 0.52 [14A]<br>0.85 [13A]<br>0.99 [12A]<br>0.33 [11A] | 4.35 ± 1.49 | 0.99 [14A]<br>0.90 [13A]<br>0.95 [12A]<br>0.99 [11A] | 0.36 ± 0.09 | <0.0001 [14A]<br>0.25 [13A]<br>0.14 [12A]<br>0.70 [11A] | 14 (8) |
| +FGF14A | -20.3 ± 4.2 | >0.99 [13A]<br>0.59 [12A]<br>0.95 [11A] | 4.47 ± 0.88 | 0.95 [13A]<br>0.86 [12A]<br>0.99 [11A] | 0.58 ± 0.12 | 0.70 [13A]<br>0.049 [12A]<br>>0.99 [11A] | 15 (10) |
| +FGF13A | -19.9 ± 5.7 | 0.81 [12A]<br>0.96 [11A] | 4.92 ± 1.87 | 0.67 [12A]<br>0.99 [11A] | 0.49 ± 0.10 | 0.99 [12A]<br>>0.99 [11A] | 5 (4) |
| +FGF12A | -16.2 ± 3.5 | 0.38 [11A] | 3.89 ± 0.59 | 0.86 [11A] | 0.45 ± 0.06 | 0.99 [11A] | 5 (1) |
| +FGF11A | -21.1 ± 6.2 | | 4.62 ± 0.96 | | 0.52 ± 0.22 | | 6 (5) |

**For GV curves, statistical analyses were performed using one-way ANOVA followed by Tukey's multiple comparisons test. For inactivation times constants, Welch's ANOVA followed by Dunnett's T3 multiple comparisons test was employed.**

**Table S2. Parameters determined from Boltzmann fits to steady-state inactivation (SSI) curves for $Na_V1.2$ alone or co-expressed with A-type FGF isoforms**

| Constructs | SSI curve | | | | |
|---|---|---|---|---|---|
| | $V_h$ (mV) | P value | z (e) | P value | N |
| $Na_V1.2$ | -62.4 ± 3.5 | <0.0001 [14A]<br><0.0001 [13A]<br>0.0001 [12A]<br>0.009 [11A] | 4.51 ± 0.50 | <0.0001 [14A]<br><0.0001 [13A]<br>0.09 [12A]<br>0.23 [11A] | 22 (16) |
| +FGF14A | -48.2 ± 2.6 | 0.94 [13A]<br>0.22 [12A]<br>0.007 [11A] | 5.85 ± 0.64 | 0.33 [13A]<br>0.001 [12A]<br><0.0001 [11A] | 31 (18) |
| +FGF13A | -49.0 ± 2.3 | 0.45 [12A]<br>0.02 [11A] | 6.11 ± 0.42 | <0.0001 [12A]<br><0.0001 [11A] | 25 (16) |
| +FGF12A | -51.8 ± 3.7 | 0.54 [11A] | 5.05 ± 0.39 | 0.98 [11A] | 8 (2) |
| +FGF11A | -55.4 ± 5.1 | | 4.91 ± 0.31 | | 11 (7) |

Statistical analyses were performed using Welch's ANOVA test followed by Dunnett's T3 multiple comparisons test (for $V_h$) or one-way ANOVA followed by Tukey's multiple comparisons test (for z).

**Table S3. Activation parameters for Na$_V$1.2_IQM expressed alone or with FGF-A isoforms.**

| Constructs | GV curve | | | | |
|---|---|---|---|---|---|
| | V$_h$ (mV) | P value | z (e) | P value | N |
| Na$_V$1.2_IQM | -23.1 ± 2.9 | 0.31 [14A] 0.99 [13A] 0.99 [12A] 0.78 [11A] | 5.01 ± 0.66 | 0.43 [14A] >0.99 [13A] >0.99 [12A] 0.06 [11A] | 12 (7) |
| +FGF14A | -26.8 ± 6.4 | 0.38 [13A] 0.49 [12A] 0.08 [11A] | 4.26 ± 0.69 | 0.34 [13A] 0.45 [12A] >0.99 [11A] | 5 (5) |
| +FGF13A | -22.5 ± 1.7 | 0.99 [12A] 0.97 [11A] | 5.13 ± 0.44 | 0.99 [12A] 0.20 [11A] | 4 (4) |
| +FGF12A | -23.4 ± 3.3 | 0.76 [11A] | 4.96 ± 0.39 | 0.049 [11A] | 7 (5) |
| +FGF11A | -21.1 ± 2.3 | | 4.39 ± 0.09 | | 6 (4) |

**Statistical analyses were performed using one-way ANOVA followed by Tukey's multiple comparisons test (for V$_h$) or Welch's ANOVA test followed by Dunnett's T3 multiple comparisons test (for z).**

**Table S4. Boltzmann fit parameters to SSI curves for $Na_V1.2\_IQM$ alone or when co-expressed with A-type FGF isoforms.**

| Constructs | SSI curve | | | | |
|---|---|---|---|---|---|
| | $V_h$ (mV) | P value | z (e) | P value | N |
| $Na_V1.2\_IQM$ | | | | | |
| +FGF14A | -44.4 ± 1.4 | 0.03 [13A]<br>0.84 [12A]<br>0.94 [11A] | 7.46 ± 1.05 | 0.30 [13A]<br>0.84 [12A]<br>0.69 [11A] | 5<br>(5) |
| +FGF13A | -40.8 ± 2.7 | 0.06 [12A]<br>0.03 [11A] | 8.64 ± 1.29 | 0.03 [12A]<br>0.02 [11A] | 6<br>(4) |
| +FGF12A | -43.5 ± 2.3 | 0.99 [11A] | 6.97 ± 1.04 | 0.99 [11A] | 10<br>(6) |
| +FGF11A | -43.8 ± 1.2 | | 6.81 ± 1.03 | | 10<br>(4) |

**Statistical analyses were performed using one-way ANOVA followed by Tukey's multiple comparisons test.**

**Table S5. Effects of FGF-A homologues on $Na_V1.2$ and
$Na_V1.2\_IQM$ GV curves, SSI and taus**

| | GV curves | | | | | | |
|---|---|---|---|---|---|---|---|
| **Constructs** | **WT** | **IQM** | | **WT** | **IQM** | | |
| | $V_h$ (mV) | | **P value** | z (e) | | **P value** | **N** |
| $Na_V1.2$ | -17.1 ± 6.5 | -23.1 ± 2.9 | 0.007 | 4.35 ± 1.49 | 5.01 ± 0.66 | 0.17 | 14/12 |
| +FGF14A | -20.3 ± 4.2 | -26.8 ± 6.4 | 0.02 | 4.47 ± 0.88 | 4.26 ± 0.69 | 0.63 | 15/5 |
| +FGF13A | -19.9 ± 5.7 | -22.5 ± 1.7 | 0.42 | 4.92 ± 1.87 | 5.13 ± 0.44 | 0.84 | 5/4 |
| +FGF12A | -16.2 ± 3.5 | -23.4 ± 3.3 | 0.004 | 3.89 ± 0.59 | 4.96 ± 0.39 | 0.003 | 5/7 |
| +FGF11A | -21.1 ± 6.2 | -21.1 ± 2.3 | 0.72 | 4.62 ± 0.96 | 4.39 ± 0.09 | 0.56 | 6/6 |
| | SSI curves | | | | | | |
| **Constructs** | **WT** | **IQM** | | **WT** | **IQM** | | |
| | $V_h$ (mV) | | **P value** | z (e) | | **P value** | **N** |
| $Na_V1.2$ | -62.4 ± 3.5 | - | | 4.51 ± 0.50 | - | | 22/na |
| +FGF14A | -48.2 ± 2.6 | -44.4 ± 1.4 | 0.003 | 5.85 ± 0.64 | 7.46 ± 1.05 | <0.0001 | 31/5 |
| +FGF13A | -49.0 ± 2.3 | -40.8 ± 2.7 | <0.0001 | 6.11 ± 0.42 | 8.64 ± 1.29 | <0.0001 | 25/6 |
| +FGF12A | -51.8 ± 3.7 | -43.5 ± 2.3 | <0.0001 | 5.05 ± 0.39 | 6.97 ± 1.04 | 0.0001 | 8/10 |
| +FGF11A | -55.4 ± 5.1 | -43.8 ± 1.2 | <0.0001 | 4.91 ± 0.31 | 6.81 ± 1.03 | <0.0001 | 11/10 |
| | Onset of Inactivation at 0 mV | | | Recovery from Inactivation at -80 | | | |
| **Constructs** | **WT** | **IQM** | | **WT** | **IQM** | | |
| | $\tau_i$ (ms) | | **P value** | $\tau_{slow}$ (ms) | | **P value** | **N** |
| $Na_V1.2$ | 0.36 ± 0.09 | - | | - | - | | 22/na |
| +FGF14A | 0.58 ± 0.12 | 0.92 ± 0.24 | 0.0004 | 592.9 ± 111.5 | 565.5 ± 88.3 | 0.64 | 31/4 |
| +FGF13A | 0.49 ± 0.10 | 1.50 ± 0.30 | 0.0002 | 2017.6 ± 324.0 | 1831.5 ± 383.5 | 0.23 | 25/6 |
| +FGF12A | 0.45 ± 0.06 | 1.33 ± 0.20 | <0.0001 | 497.2 ± 97.7 | 565.0 ± 112.5 | 0.20 | 8/10 |
| +FGF11A | 0.52 ± 0.22 | 0.88 ± 0.16 | 0.009 | 173.1 ± 19.7 | 155.8 ± 34.5 | 0.17 | 11/10 |

**Statistical analyses between the two groups (WT vs IQM)
were performed using unpaired t test.**

**Table S6. Use-dependent changes in FGF-A-mediated recovery from LTI**

| 10-Pulse Train frequencies | w/o FGF-A | | | | | |
|---|---|---|---|---|---|---|
| | $\tau_f$ (ms) | $A_f$ | $\tau_s$ (ms) | $A_s$ | | N |
| 1P | 8.67 ± 2.27 | 0.99 ± 0.02 | - | - | | 22/0 |
| 5 Hz | 8.27 ± 3.24 | 0.95 ± 0.03 | 97.3 ± 41.0 | 0.06 ± 0.01 | | 3/6 |
| 10 Hz | 8.99 ± 3.19 | 0.94 ± 0.03 | 153.0 ± 59.8 | 0.05 ± 0.03 | | 3/6 |
| 20 Hz | 9.29 ± 2.89 | 0.91 ± 0.05 | 105.6 ± 39.2 | 0.12 ± 0.03 | | 3/6 |
| 40 Hz | 10.23 ± 3.67 | 0.90 ± 0.05 | 131.2 ± 29.4 | 0.09 ± 0.03 | | 4/5 |
| | + FGF14A | | | | | |
| 1P | 5.38 ± 1.79 | 0.40 ± 0.06 | 592.9 ± 111.5 | 0.60 ± 0.06 | | 31 |
| 2 Hz | 6.13 ± 2.91 | 0.23 ± 0.05 | 648.6 ± 34.0 | 0.74 ± 0.08 | | 2 |
| 5 Hz | 6.62 ± 2.16 | 0.16 ± 0.04 | 662.9 ± 83.9 | 0.83 ± 0.04 | | 9 |
| 10 Hz | 8.66 ± 2.38 | 0.11 ± 0.04 | 629.1 ± 78.3 | 0.86 ± 0.03 | | 8 |
| 20 Hz | 9.49 ± 3.39 | 0.09 ± 0.03 | 620.4 ± 70.3 | 0.91 ± 0.06 | | 8 |
| 40 Hz | 12.07 ± 8.22 | 0.07 ± 0.04 | 635.1 ± 68.7 | 0.91 ± 0.04 | | 8 |
| | + FGF13A | | | | | |
| 1P | 6.31 ± 1.74 | 0.54 ± 0.06 | 2017.6 ± 324.0 | 0.45 ± 0.06 | | 25 |
| 5 Hz | 15.33 ± 12.66 | 0.17 ± 0.06 | 2426.2 ± 394.2 | 0.79 ± 0.06 | | 5 |
| 10 Hz | 5.66 | 0.14 | 1614 | 0.77 | | 1 |
| 20 Hz | 9.05 ± 3.34 | 0.15 ± 0.08 | 1900.7 ± 629.5 | 0.79 ± 0.11 | | 3 |
| 40 Hz | 14.75 ± 9.26 | 0.10 ± 0.04 | 2230.8 ± 524.5 | 0.86 ± 0.05 | | 5 |
| | + FGF12A | | | | | |
| 1P | 4.96 ± 2.60 | 0.70 ± 0.04 | 497.2 ± 97.7 | 0.29 ± 0.04 | | 8 |
| 5 Hz | 4.69 ± 0.64 | 0.44 ± 0.07 | 633.5 ± 52.4 | 0.54 ± 0.07 | | 5 |
| 10 Hz | 5.42 ± 0.84 | 0.32 ± 0.08 | 602.2 ± 44.5 | 0.64 ± 0.08 | | 5 |
| 20 Hz | 7.25 ± 2.25 | 0.24 ± 0.09 | 592.2 ± 81.6 | 0.74 ± 0.09 | | 5 |
| 40 Hz | 8.64 ± 1.72 | 0.19 ± 0.10 | 540.5 ± 53.6 | 0.77 ± 0.10 | | 5 |
| | + FGF11A | | | | | |
| 1P | 7.92 ± 2.73 | 0.55 ± 0.05 | 173.1 ± 19.7 | 0.44 ± 0.05 | | 11 |
| 5 Hz | 8.28 ± 2.39 | 0.45 ± 0.04 | 230.8 ± 39.2 | 0.51 ± 0.04 | | 5 |
| 10 Hz | 9.61 ± 2.02 | 0.39 ± 0.05 | 239.0 ± 19.3 | 0.58 ± 0.04 | | 5 |
| 20 Hz | 10.04 ± 3.02 | 0.32 ± 0.04 | 227.2 ± 11.7 | 0.65 ± 0.05 | | 5 |
| 40 Hz | 10.93 ± 5.85 | 0.29 ± 0.03 | 201.0 ± 56.3 | 0.68 ± 0.04 | | 5 |