## [Peer Review File · The Journal of General Physiology]

Long Term Inactivation mediated by different FGF-A homologues on heterologously expressed NaV1.2 currents

Yenisleidy Lorenzo-Ceballos, Pedro Martinez-Espinosa, Xiao-Ming Xia, and Christopher Lingle

Corresponding Author(s): Christopher Lingle, Washington University in St. Louis School of Medicine

Review Timeline:

Submission Date:	February 18, 2026
Editorial Decision:	March 18, 2026
Revision Received:	March 20, 2026
Editorial Decision:	March 23, 2026
Revision Received:	March 30, 2026

Editor: Olaf Andersen

Transaction Report:

DOI: <https://doi.org/10.1085/jgp.202613985>

March 18, 2026

Dr. Christopher J Lingle
Washington University in St. Louis School of Medicine
Anesthesiology
Box 8054
St. Louis, MO 63110

Re: 202613985

Dear Chris,

Thank you for submitting your manuscript, entitled "Comparison of Long Term Inactivation mediated by different FGF-A homologues on heterologously expressed NaV1.2 currents" to JGP. Your manuscript has now been seen by 3 reviewers, whose comments are appended below. You will see that the reviewers were enthusiastic about the study and its potential impact and raised only minor concerns, which should be addressed prior to further consideration of the manuscript at JGP. In particular, please view the editor's summary directly below.

Editor's Summary

Whereas Reviewer #2 has no suggestions for improving/revising the presentation, Reviewers #1 and #3 identify areas where the presentation could be improved/tightened. The suggestions call for a thorough, straightforward revision, which should cause no problems. Based on my own reading, I think you should (re)define FGF in line 95, and you should reconsider the wording on line 548; do the N-termini "support" right-shifted SSI curves or do they "shift the curves to the right"? I also believe you should remove "Comparison of" in the title; yes, you do compare, but you go far beyond the comparison.

We hope that you will be able to submit a revised manuscript that addresses these points, which we believe will pose no problems, and which may be re-reviewed. In addition, please do not hesitate to contact me (via the editorial office) if you feel that a discussion of the reviewers' and editors' comments would be helpful.

Please submit your revised manuscript via the link below, along with a point-by-point letter that details your response to the reviewers' and editor's summary, as well as a copy of the text with alterations highlighted (boldfaced or underlined). If the article is eventually accepted, it would include a 'revised date' as well as submitted and accepted dates. If we do not receive the revised manuscript within one year, we will regard the article as having been withdrawn. We would be willing to receive a revision of the manuscript at a later time, but the manuscript will then be treated as a new submission, with a new manuscript number.

Please pay particular attention to recent changes to our instructions to authors in the following sections: Data presentation, Blinding and randomization and Statistical analysis, under Materials and Methods, as shown here: <https://rupress.org/jgp/pages/submission-guidelines#prepare>. Re-review will be contingent on inclusion of the required information (including for data added during revision) and demonstration of the experimental reproducibility of the results. Also, To improve the reproducibility of published content, we have partnered with SciScore. Authors are prompted in eJP to copy and paste the Materials and Methods section of their manuscript for a SciScore assessment when submitting their revised manuscript. Authors are encouraged (not required) to further revise their Materials and Methods if the SciScore is below 4. More information can be found here: <https://rupress.org/jgp/pages/submission-guidelines#sciscore>.

Please note, JGP now requires authors to submit Source Data used to generate figures containing gels and Western blots with all revised manuscripts (when applicable). This Source Data consists of fully uncropped and unprocessed images for each gel/blot displayed in the main and supplemental figures. If your paper includes cropped gel and/or blot images, please be sure to provide one Source Data file for each figure that contains gels and/or blots along with your revised manuscript files. File names for Source Data figures should be alphanumeric without any spaces or special characters (i.e., SourceDataF#, where F# refers to the associated main figure number or SourceDataFS# for those associated with Supplementary figures). The lanes of the gels/blots should be labeled as they are in the associated figure, the place where cropping was applied should be marked (with a box), and molecular weight/size standards should be labeled wherever possible. Source Data files will be made available to reviewers during evaluation of revised manuscripts and, if your paper is eventually published in JGP, the files will be directly linked to specific figures in the published article.

Source Data Figures should be provided as individual PDF files (one file per figure). Authors should endeavor to retain a minimum resolution of 300 dpi or pixels per inch. Please review our instructions for export from Photoshop, Illustrator, and PowerPoint here: <https://rupress.org/jgp/pages/submission-guidelines#revised>

Whilst you are revising your manuscript, we ask that you consider whether you have any artwork that might be suitable for the

cover of JGP. Microscopy images are particularly good for cover artwork, but other types of image can be very effective, so we encourage you to be creative. Please don't restrict yourself to images from the paper; an image that is relevant to the work described would be just as suitable. Images should be a minimum resolution of 300 dpi. To see recent examples, visit the following page and click on 'Show covers? Yes': <https://jgp.rupress.org/content/by/year>)

Thank you for submitting your interesting research to JGP.

Please submit your revised manuscript, and any associated files, via this link:
Link Not Available

Sincerely,

Olaf
Olaf S. Andersen, M.D.
On behalf of Journal of General Physiology

Journal of General Physiology's mission is to publish mechanistic and quantitative molecular and cellular physiology of the highest quality; to provide a best-in-class author experience; and to nurture future generations of independent researchers.

Reviewer #1 (Comments to the Authors):

The manuscript by Lorenzo-Ceballos from the Lingle lab present a comprehensive evaluation of fast and long-term inactivation (LTI) exhibited by Nav1.2 channels in the absence and presence of cytoplasmic fibroblast growth factor homologous proteins (FHF and other names), specifically the A-isoforms (distinguished by unique N-terminal sequences). LTI evoked by FHF is likely important physiologically for fine tuning Nav channel availability. This study demonstrated notable functional differences among 4 isoforms (11A-14A) and suggest that these differences arise from allosteric mechanisms (supported by data and a new kinetic model). The paper is data rich, rigorous and well written. Overall, the conclusions are well support by the data. There are some concerns that should be addressed.

Major concerns

1. LTI seems mostly distinguished by a prominent slow component of recovery from inactivation evoked by a brief (10 ms) depolarizing prepulse (Fig. 3) and by depolarized steady-state inactivation evoked by 500 ms prepulses (Fig. 2). Nav channels can also exhibit slow inactivation typically evoked by very long depolarizing prepulses (> 5 sec) and revealed after a brief hyperpolarizing pulse to allow channels to escape from fast inactivation. Some consideration should be given to the potential contamination of LTI by slow inactivation, which might be partially evoked during the 500 ms prepulse. Also, brief discussion of slow inactivation as a phenomenon in the background section of the paper to frame the LTI story would be helpful. If effects of FHF on slow inactivation has not been studied previously, then this should be discussed (maybe future work?).
2. Multiple terms were used interchangeably to describe fast inactivation ('traditional', 'intrinsic', 'normal', IF) and this is confusing. Best to simplify and be consistent with such terms.
3. Line 431 implies that the FGF-A protein 'dissociates' at a distinct rate, but the experiments do not measure dissociation. Similarly, lines 485, 486, 515, 517, and possibly elsewhere, the authors imply 'binding' rate, which are not directly measured. Perhaps calling these apparent binding rates is OK, but might be best to avoid implying binding at all.
4. Specific classes of neurons are likely to express more than one FHF isoform and the net effects of each cell's unique portfolio on Nav channel availability is difficult to predict. Can the authors offer an approach to dissect this, perhaps using computational modeling?

Minor comments

Line 147 - 'fector' should be 'vector'
Line 917 - word 'spontaneous' is used twice

Reviewer #2 (Comments to the Authors):

Lorenzo-Ceballos et al. have done an extensive analysis of Nav1.2 inactivation in the absence or presence of intracellular FGF (FHF) isoforms. Paralleling prior studies on Nav1.6, they find that A-isoforms of FGF/FHF induce long-term inactivation of Nav1.2, and that this requires amino acids in the previously characterized long-term inactivation particle located at the N-terminus that is similar among A-isoforms, but absent from other FGF/FHF isoforms. An interesting finding is that the A-isoforms

differ in the degree to which they can induce Nav1.2 long-term inactivation, with FGF13/FHF2 being most potent, and FGF11/FHF3 being the least. They further show that the various A-isoforms similarly differ in their potency of long-term inactivation when assayed on fast inactivation-defective Nav1.2(IQM), thereby showing that FGF/FHF A-isoforms substantially differ in their ability to induce and sustain long-term inactivation. These experiments were rigorously conducted and analyzed.

Reviewer #3 (Comments to the Authors):

This is a fascinating and carefully executed study examining long-term inactivation (LTI) of neuronal Nav1.2 channels produced by co-expression of four FGF isoforms. The first described mechanism of fast inactivation (FI) in Nav channels occurs rapidly with membrane depolarization and the channel also recovers rapidly. Although FI was originally conceived to occur through a ball-and-chain mechanism, recent evidence suggests that FI involves closure of the internal pore. Co-expression of Nav channels with soluble FGFs induces long-term inactivation that occurs with similar fast kinetics but where recovery from inactivation is much slower. Thus, this diversity of inactivation mechanisms provides the means of tuning inactivation behavior to provide excitable cells with short term memory of their prior activity with variable longevity. Here the authors systematically measure the rates of onset and recovery from inactivation for Nav1.2 in the absence and presence of four FGFs, as well as selected mutants in FGF-14A to arrive at several important conclusions. None of the FGFs appreciably alter voltage-dependent activation of Nav1.2, but detectably slow fast inactivation, shift the steady-state inactivation curve to depolarized voltages while dramatically slowing recovery from inactivation with rates that vary about an order of magnitude. All of these measurements are repeated in the IQM mutant background to clearly distinguish effects of FGFs on LTI in the absence of conventional fast inactivation. Most of the selected mutants in the N-terminus of FGF-14A completely prevented the development of LTI, implicating this region as critical for the mechanism of LTI. The study concludes by exploring the availability of Nav 1.2 during trains of depolarizations with both FI and LTI present with different FGFs or with FI removed, demonstrating how different FGFs play a role in tuning cells to particular firing frequencies. The author also discuss models (Scheme 2 and 2b) wherein both FI and LTI occur from the open state and evaluate the extent to which the models follow the experimental data. The authors also present evidence throughout the study to support the conclusion that the effects of FGFs on FI and steady-state inactivation occur through a mechanism distinct from LTI and likely not involving the N-terminus. Overall, I found this to be a meticulously constructed study with thoughtfully conceived experimental protocols, extremely high-quality data, and rigorously constructed conclusions. I have only minor suggestions for the authors, and they are to be congratulated on a beautiful body of work.

1) Line 391 I am curious why such long 500 ms steps were used to generate steady-state inactivation curves when both FI and LTI inactivation occur on much shorter timescales. Might an effect of FGFs on slow inactivation explain why they shift the steady-state inactivation relationship in a way that is not explained by LTI? Perhaps something to comment on in the discussion. Line 436 Maybe change 'unblocking' to 'recovery' since the pore blocking mechanism for FGFs is only brought into the story in the discussion.

2) Line 449 I suggest inserting WT before Nav1.2

3) I could not find the protocol details for the results presented in Fig. 5

4) Line 877 needs to insert 'the C-terminal domain of' before Nav1.5

Responses to suggestions of the editor and reviewers

First of all, we thank the reviewers and editor for their supportive comments regarding the manuscript.

In accordance with the suggestions of the editor, we made the recommended changes to the title, redefining FGF for clarity at line 95 and the wording in line 548.

Review #1.

Major Concerns:

1. Both Rev. #1 and #3 raise questions pertinent to slow inactivation of NaV channels. One concern is whether some aspects of our results might be contaminated by slow inactivation. Another potential concern not directly raised by the reviewers, but could be a concern for readers, is whether LTI might in some way be related to slow inactivation.

In our view, slow inactivation remains a somewhat murky topic. In fact, there really are no systematic studies that have evaluating slow inactivation for different NaV homologues with a standard set of protocols. Examination of different papers suggests that different NaV homologues, including Nav1.2, 1.3, 1.5 and 1.6 may exhibit major differences in the extent to which a 500 ms depolarization to 0 mV will result in slow inactivation. In our work with Nav1.3 in chromaffin cells, we explicitly looked for slow recovery components after 500 ms depolarizations to assess potential contamination by slow inactivation and that 500 ms steps resulted in only about 10% of channels entering slow inactivated states. In this earlier work, we chose 500 ms conditioning steps for SSI curves, because it was only after 500 ms that the SSI curves coincided at a limiting $V_{0.5}$ of availability. Because of the slow recovery from LTI, it takes hundreds of ms to achieve an approach to steady-state in availability.

With Nav1.2, we have not done a full set of tests of depolarizations of different durations to assess the extent of development of slow inactivation. However, we note that Figure 10B indicates that during a 40 Hz train of 10 steps to 0 mV with WT Nav1.2, there is only minor suggestion of a slow component of recovery from inactivation. As such we do not think that slow inactivation impacts on any of measurements pertinent to use-dependent changes in the time constants and relative amplitude of fast and slow recovery components for the different Nav+FGF-A combinations. We cannot fully exclude that, in the SSI, protocol there might be some small component of entry into slow inactivation. However, we think that does not impact on the reported FGF-A homologue induced changes in the $V_{0.5}$ for SSI.

Concerning the question of whether LTI might reflect a slow inactivation process dramatically altered in an allosteric fashion by an FGF-A, the short answer is that this seems highly unlikely, although it may not be possible to rigorously exclude it. Although both slow inactivation and LTI exhibit a slow recovery behavior (differing in over an order of magnitude or more in some cases), the rapid onset of LTI, that it can readily produce full inactivation (in the absence of intrinsic fast inactivation), and that specific residues in the FGF-A N-terminal inactivation segment are responsible for inactivation all support the idea that LTI is a completely unrelated mechanism from slow inactivation. As with slow inactivation and fast inactivation, slow inactivation is likely to occur in parallel with LTI and not compete with it.

To help readers better evaluate LTI within the context of slow inactivation, we now include the following passage in the Introduction: **Given the slow recovery from inactivation characteristic of LTI, it is important to emphasize that LTI is considered a mechanistically distinct process from the intrinsic slow inactivation observed for most Na_v channels (Goldfarb, 2012). The primary difference is that LTI is a rapid onset process occurring within ms, whereas slow inactivation is of slow onset developing over 100s of milliseconds or even seconds to minutes (Ulbricht, 2005).**

In the methods, we have added:

“For steady-state inactivation curves (SSI), we utilized 500 ms prepulses to voltages from -120 mV to 0 mV. In our previous work with native Na_v currents in rat (Martinez-Espinosa et al., 2021a) and mouse (Martinez-Espinosa et al., 2021b) chromaffin cells, we noted that 500 ms prepulses were required in order that the SSI curves reached a limiting V_h value. Such currents primarily arise from $\text{Na}_v1.3+\text{FGF14A}$. In such experiments we found that a 500 ms depolarization to 0 mV resulted in an increase in the slow recovery component that could be attributed to about 10% of the channels entering slow inactivation (Martinez-Espinosa et al., 2021a). However, other results suggest that there may be variability in both the rapidity and extent of development of slow inactivation among different Na_v homologues (Vilin et al., 2012; Abdelsayed et al., 2013; Zhang et al., 2013). In our own hands, 100 ms depolarizations to 0 mV with WT $\text{Na}_v1.2$ produces minimal slow recovery, similar to another recent study on human $\text{Na}_v1.2$ (Ganguly et al., 2021). Furthermore, in Figure 10B, it can be seen that, for WT $\text{Na}_v1.2$ currents, a 40 Hz train of 10 pulses results in minimal development of any component of slow recovery from inactivation. Thus, the repetitive stimulation protocols used in the present work are unlikely to produce appreciable slow inactivation. However, we note that, in contrast to the results with human $\text{Na}_v1.2$, for rat $\text{Na}_v1.2$, a 500 ms depolarization

to 0 mV resulted in about ~40% of the channels entering a slow inactivated condition (Abdelsayed et al., 2013). In the present experiments, it must be considered possible that the SSI curves may be impacted to some extent at the more positive prepulses by some entry into slow inactivated states. However, such an effect seems unlikely to impact on any conclusions pertinent to the effects of different FGF-A homologues on the SSI properties.”

Finally, in the Discussion we add the following.

“Might slow inactivation impact on any of our results and interpretations. In the results, we acknowledged that the 500 ms prepulse used in our SSI protocols may have resulted in some slow inactivation, but that it was unlikely to impact on our observations regarding shifts in SSI curves mediated by the FGF-A homologues. Furthermore, Figure 10B shows that even a 10 pulse 40 Hz train of steps to 0 mV produces minimal entry into slow inactivation in WT Na_v1.2 channels. Thus, LTI occurs under conditions for which slow inactivation does not occur. Given the rapidity of the LTI inactivation onset, that it results in essentially full inactivation of Na_v1.2 when intrinsic fast inactivation removed, and that it involves specific residues in the FGF-A N-terminal segment, the idea that LTI might arise from alteration of the intrinsic slow inactivation process seems unlikely. Although future experiments would be required to fully address this topic, we propose that, as is the case for intrinsic fast and slow inactivation, the occurrence of slow inactivation will occur in parallel with LTI. “

2. We have now chosen “intrinsic fast inactivation” for all cases, except when I_F is used in relationship to kinetic schemes.
3. The reviewer is correct that we are not explicitly measuring dissociation or association. We have therefore modified the terminology so that, up until the section of the paper when we look at N-terminal mutations, we only refer to onset on inhibition or inactivation and recovery. Near the beginning of the section addressing the mutational evaluations we add the following text.

Furthermore, such results have led to the proposal that the rate of onset of LTI likely reflects the association of the N-terminal inactivation domain with a position that inhibits ion flux, while recovery from inactivation reflects dissociation of the blocking particle from its site of occupancy (Dover et al., 2010) . Whether such a simple occlusion model applies to LTI has not been fully established, but nothing as yet excludes that possibility. Within this context, it is convenient to consider that the forward rate of inactivation reflects a molecular association process, while recovery may reflect dissociation.

Whether such a simple occlusion model applies to LTI has not been fully established, but nothing as yet excludes that possibility. Within this context, it is convenient to consider that the forward rate of inactivation reflects a molecular association process, while recovery may reflect dissociation. As such, we feel it is important and useful that the idea that the onset of LTI reflects movement of the LTI particle into its blocking position (association) and that recovery from LTI reflects dissociation of that particle from its position of inactivation be emphasized. We feel that an important aspect of the present paper is that the time of recovery from inactivation, which we attribute to dissociations, can be measured robustly, is unaffected by the presence or absence of fast inactivation, and can be altered by N-terminal mutations. This potentially provides a basis upon which meaningful manipulations of compounds that might manipulate LTI can be evaluated. Therefore, considering the recovery rate in terms of a dissociation process has utility in guiding future work and interpretation.

4. The reviewer raises an interesting question pertinent to the likely coexpression of multiple FHF/FGF isoforms in specific classes of neurons. The problem is made more complicated by the likelihood that specific NaVs and specific FGFs may be distributed to different loci in a cell, and then there is also the issue of which N-terminal variants (e.g., A vs. B) of a given FGF may be present. Given our interest specifically in the impact of LTI, a concern of this lab is that it remains not at all clear in what native cells FGF-A homologues may actually be expressed, as we mention in the Discussion. Antibodies and RNA have not yet seemed adequate for this purpose. However, we do think that better selection of protocols by investigators to probe the properties of any recovery from LTI would help reveal how pervasive FGF-A mediated LTI may be.

Minor comments were addressed as recommended.

Reviewer #2. We thank the reviewer for the supportive comments.

Reviewer #3. Thank you for the kind remarks. Very much appreciated.

1. The issues pertinent to steady-state inactivation were addressed in response to Rev. #1.
2. Also in our response to Rev. 1, we made changes in the wording regarding unblocking vs. recovery, up until the point where it becomes useful to further discuss the results in terms of a pore-blocking mechanism.
3. WT was added to present NaV1.2 in multiple locations.

4. The legend of Figure 5 now indicates that the steady-state inactivation protocols were identical to those in Figure 2.
5. 5. Change made.

Dr. Christopher J Lingle
Washington University in St. Louis School of Medicine
Anesthesiology
Box 8054
St. Louis, MO 63110

Re: 202613985R1

Dear Chris,

I am pleased to let you know that your manuscript, titled "Long Term Inactivation mediated by different FGF-A homologues on heterologously expressed NaV1.2 currents" is scientifically acceptable for publication in Journal of General Physiology. Formal acceptance will follow when it is modified in accordance with our editorial policies (see below).

Please note items that need attention are listed at the bottom of this email (under 'manuscript formatting checklist'). Your manuscript should be a double-spaced MS Word file and include editable tables, if appropriate.

Lastly, JGP requires a data availability statement for all research article submissions. These statements will be published in the article directly above the Acknowledgments. The statement should address all data underlying the research presented in the manuscript. Please visit the JGP instructions for authors for guidelines and examples of statements at <https://rupress.org/jgp/pages/editorial-policies#data-availability-statement>.

Please submit your final files via this link:
Link Not Available

Thank you for choosing to publish your research in JGP and please feel free to contact me with any questions.

Sincerely,

Olaf
Olaf S. Andersen, M.D.
On behalf of Journal of General Physiology

Journal of General Physiology's mission is to publish mechanistic and quantitative molecular and cellular physiology of the highest quality; to provide a best in class author experience; and to nurture future generations of independent researchers.

Manuscript formatting checklist:

- MS Word document of text needed (including editable tables)
- MS Word document of supplemental text needed, if applicable (including figure legends and editable tables)
- Brief Statement describing supplementary information needed, if applicable (in subsection at end of Materials & Methods)
- Please include a data availability statement preceding the Acknowledgments section. Please see <https://rupress.org/jgp/pages/editorial-policies#data-availability-statement>
- Figures created at sufficient resolution and in acceptable format (including supplemental if applicable). If working in Illustrator, we prefer .ai or .eps file format. If working in Photoshop please use 600dpi/1000dpi .tiff or .psd file format. Minimum resolution at estimated print size: Minimum resolution for all figures is 600 dpi. For figures that contain both photographs and line art or text, 600 dpi is highly recommended. Figures containing only black and white elements (line art, no color, and no gray) should be 1,000 dpi. Maximum figure size is 7 in wide x 9 in high (17.5 x 22.8 cm) at the correct resolution. <https://jgp.rupress.org/fig-vid-guidelines>
- Supplemental figures, if any, conforming to same guidelines as manuscript figures (noted above)
- If images resemble one from a prior publications, the author must seek permissions (to reproduce or adapt) from the original publisher. [You can resubmit your paper while waiting to hear back from the original publisher but please keep us updated]
- All authors must complete a disclosure form prior to acceptance. A link to complete the form has been sent to all coauthors. Please provide the editorial office with updated email addresses if necessary